

# The Laurentide Ice Sheet in southern New England and New York during and at the end of the Last Glacial Maximum - A cosmogenic-nuclide chronology

Allie Balter-Kennedy[1,2], Joerg M. Schaefer[1,2], Greg Balco[3], Meredith A. Kelly[4], Michael R. Kaplan[1], Roseanne Schwartz[1], Bryan Oakley[5], Nicolás E. Young[1], Jean Hanley[1], Arianna M. Varuolo-Clarke[1,2]

[1]Lamont–Doherty Earth Observatory, Columbia University, Palisades, NY 10964, USA

[2]Department of Earth and Environmental Sciences, Columbia University, New York, NY 10027, USA

[3]Berkeley Geochronology Center, Berkeley, CA 94709 USA.

[4]Department of Earth Sciences, Dartmouth College, Hanover, NH 03755, USA

[5]Environmental Earth Science Department, Eastern Connecticut State University, Willimantic, CT, 06226, USA

*Correspondence to*: Allie Balter-Kennedy (abalter@ldeo.columbia.edu)

**Abstract.** We present 40 new [10]Be exposure ages of moraines and other glacial deposits left behind by the southeastern sector of the Laurentide Ice Sheet (LIS) in southern New England and New York, summarize the regional moraine record, and interpret the dataset in the context of previously published deglaciation chronologies. The regional moraine record spans the Last Glacial Maximum (LGM), with the outermost ridge of the terminal complex dating to ~26–25 ka, the innermost ridge of the terminal complex dating to ~22 ka, and a series of smaller recessional limits within ~50 km of the terminal complex dating to ~21–20.5 ka. The chronology generally agrees with independent age constraints from radiocarbon and glacial varves. A few inconsistencies among ages from cosmogenic-nuclide measurements and those from other dating methods are explained by geologic scatter where several bedrock samples and boulders from the outer terminal moraine exhibit nuclide inheritance, while exposure ages on large moraines are likely affected by postdepositional disturbance. The exposure-age chronology places the southeastern sector of the LIS at or near its maximum extent from ~26 to 21 ka, which is broadly consistent with the LGM sea-level lowstand, local and regional temperature indicators, and local summer insolation. The net change in LIS extent represented by this chronology occurred more slowly (<5 to 25 m yr$^{-1}$) than retreat through the rest of New England, consistent with a slow general rise in insolation and modeled summer temperature. We conclude that the major pulse of LIS deglaciation and accelerated recession, recorded by dated glacial deposits north of the moraines discussed here, did not begin until after atmospheric $CO_2$ increased at ~18 ka, marking the onset of Termination 1.

**Short Summary.** We date sedimentary deposits indicating the southeastern Laurentide Ice Sheet was at or near its southernmost extent from ~26,000 to 21,000 years ago when sea-level was lowest and other climate records indicate glacial conditions. Slow deglaciation began ~22,000 years ago alongside a slow but steady rise in modeled local



summer temperature, but significant deglaciation in the region did not begin until ~18,000 years ago when atmospheric
CO$_2$ began to rise, signaling the end of the last ice age.

## 1 Introduction

We describe new cosmogenic-nuclide exposure ages on moraines and other glacial-margin deposits in

southern New England and New York that track the timing and position of the margin of the southeastern sector of
the Laurentide Ice Sheet (LIS) during the Last Glacial Maximum (LGM; 26.5–19 ka) and Termination 1 (18–11 ka),
the most recent glacial-interglacial transition. The LIS held ~50–80 m sea-level equivalent at its greatest extent during
the LGM (Clark et al., 2009, 1996; Denton and Hughes, 1981; Stokes, 2017; Stokes et al., 2012), making it the largest
ice sheet of the last glacial period, and then deglaciated as temperature and CO$_2$ returned to interglacial values during
Termination 1 (Broecker and Donk, 1970; Cuffey et al., 2016; Dalton et al., 2020; Denton et al., 2010; Dyke, 2004;
Marcott et al., 2014). LIS topography, albedo, and meltwater exerted major forcing on large-scale atmospheric
dynamics (Löfverström et al., 2014; Ullman et al., 2014), ocean circulation (Clark et al., 2001; Denton et al., 2010;
McManus et al., 2004), and sea-level (Clark et al., 2009; Lambeck et al., 2014; Stokes, 2017) during the LGM and
subsequent deglaciation. We focus on the southeastern sector of the LIS, which is particularly important because of
its proximity to the North Atlantic Ocean, meaning that meltwater from this sector had the potential to suppress the
Atlantic Meridional Overturning Circulation (AMOC), inducing global-scale climate feedbacks (Barker et al., 2009;
Barker and Knorr, 2021; Buizert et al., 2014; Denton et al., 2010; McManus et al., 2004). Improving LIS chronologies
bears on better understanding of regional paleoenvironmental and paleoclimatic changes.

Cosmogenic-nuclide and radiocarbon dating have been used to show that the LIS fluctuated at or near its full

LGM extent until ~22 ka, with terminal moraines dating to ~23–22 ka in the midwestern United States (Curry and
Petras, 2011; Glover et al., 2011; Heath et al., 2018; Ullman et al., 2015) and to ~26–24 ka in the northeastern United
States (Balco et al., 2002; Balco and Schaefer, 2006; Corbett et al., 2017; Stanford et al., 2021). Margin retreat
potentially accelerated across the LIS by ~20.5 ka (Balco and Schaefer, 2006; Ullman et al., 2015). Therefore, the
initial retreat of the LIS margin from its LGM limits coincided with a steady rise in boreal summer insolation that
began ~24 ka (Clark et al., 2009; Denton et al., 2010; Hays et al., 1976; Milankovitch, 1941; Ullman et al., 2015), and
began several thousand years before the deglacial rise in CO$_2$ observed in the Antarctic ice core record (Marcott et al.,
2014). The LIS might have been sensitive to this relatively weak orbital forcing in its full glacial configuration,
although orbital forcing alone was likely insufficient to force the return to full interglacial conditions (Barker and
Knorr, 2021; Denton et al., 2010; Imbrie et al., 1993; Raymo, 1997; Tzedakis et al., 2018). The increase in atmospheric
CO$_2$ beginning around 18 ka likely played a key role in the full deglaciation of the LIS (Gregoire et al., 2015; Marcott
et al., 2014; Shakun et al., 2015).

Prominent moraines in northern New Jersey, and coastal New York and New England, along with a series of

smaller recessional moraines, mark the LIS extent during the LGM and afford an opportunity to constrain the timing
of the LGM and initial deglaciation during Termination 1. These moraines are morphostratigrahically correlated across
the region and bracketing radiocarbon ages from a few locations have been used to estimate the ages for the entire
moraine sequence (Stone and Borns, 1986; Stone et al., 2005). Several of the moraine segments have now also been



dated using cosmogenic nuclides (Balco et al., 2009, 2002; Balco and Schaefer, 2006; Corbett et al., 2017). Our 40
new [10]Be ages from Rhode Island, Long Island, New York City, and the Lower Hudson Valley complement existing
moraine chronologies for the LIS margin and, together, these chronologies provide net changes in LIS extent as well
as retreat rate estimates for this climatically important sector.

**1.1 Existing LIS chronologies in southern New England, New York, and northern New Jersey**

**1.1.1 Regional moraine stratigraphy**

Regional LIS margin positions have been inferred across the northeastern United States using various glacial
deposits, including moraines, glacial lake sediment, ice-contact deltas, and morphosequences of contemporaneous ice-
marginal to -distal landforms and sediment facies (e.g., Cadwell, 1989; Fuller, 1914; Koteff and Pessl, 1981;
McMaster, 1960; Stone and Borns, 1986; Stone et al., 2005; Woodworth and Wigglesworth, 1934). Importantly, these
deposits mark the most recent extension of the ice margin to a given position because each advance of the ice sheet
removes evidence of previous ice-margin fluctuations. The most prominent of these features is a terminal moraine
complex that defines the modern coastline of New England and New York, composed of two massive end moraine
systems that were constructed during the most extensive LGM advances of the Hudson, Connecticut, and
Narragansett-Buzzards Bay lobes. These large moraines (50–100 m tall, 2–10 km wide) are characterized by
imbricated thrust sheets of outwash deposits and dislocated preglacial sediment displaced during ice-margin advance
and are overlain by till in many places (Fuller, 1914; Kaye, 1972, 1964a, 1964b; Mills and Wells, 1974; Oldale and
O'Hara, 1984; Sirkin, 1982; Boothroyd and Sirkin, 2002). Crosscutting relationships among segments within the
moraine systems and, importantly, the glaciotectonic nature of the deposits combined with the presence of overlying
till suggest that the moraines were formed during phases of ice-margin advance as the LIS fluctuated at or near its
southernmost reaches during the last glaciation (Oldale and O'Hara, 1984; Sirkin, 1976; Boothroyd and Sirkin, 2002).
The outermost component of the terminal complex can be traced from the Budd Lake moraine in northern New Jersey,
to the Harbor Hill and Ronkonkoma moraines on Long Island, New York  and across Block Island Sound to Martha's
Vineyard and Nantucket (Figure 1; Stone and Borns, 1986). About 10–30 km north of the outer terminal limit, the
innermost element of the terminal moraine complex records the last major LGM ice advance in the region and is
correlated across Long Island's north shore to Fisher's Island, Connecticut, the Charlestown moraine in Rhode Island,
and the Buzzards Bay moraine on Cape Cod (Figure 1; Sirkin, 1976; Sirkin, 1982; Stone and Borns, 1986).
A series of ice-contact deltas has been used to correlate the ice-margin position along Long Island's north
shore across New York City to the Ogdensburg-Culvers Gap moraine in northern New Jersey (Figure 1; Stanford,
1993; Stanford et al., 2021; Stone et al., 1995, 2005). The easternmost of these deltas in lower Manhattan is associated
with glacial Lake Bayonne, the presence of which required that the ice margin was located at or south of the Sands
Point moraine on Long Island to block a spillway at Hell Gate (Figure 1; Stanford et al., 2021; Stanford and Harper,
1991; Stone et al., 2005). The large coastal moraines dammed lakes fed by LIS meltwater as the ice-margin retreated
northward, and the associated lakefloor deposits are found throughout northern New Jersey (Stanford et al., 2021) and
underlie much of what is now Long Island Sound (Stone et al., 2005), Narragansett Bay (Oakley, 2012), Block Island
Sound, and Rhode Island Sound (Needell et al., 1983; Frankel and Thomas, 1966).





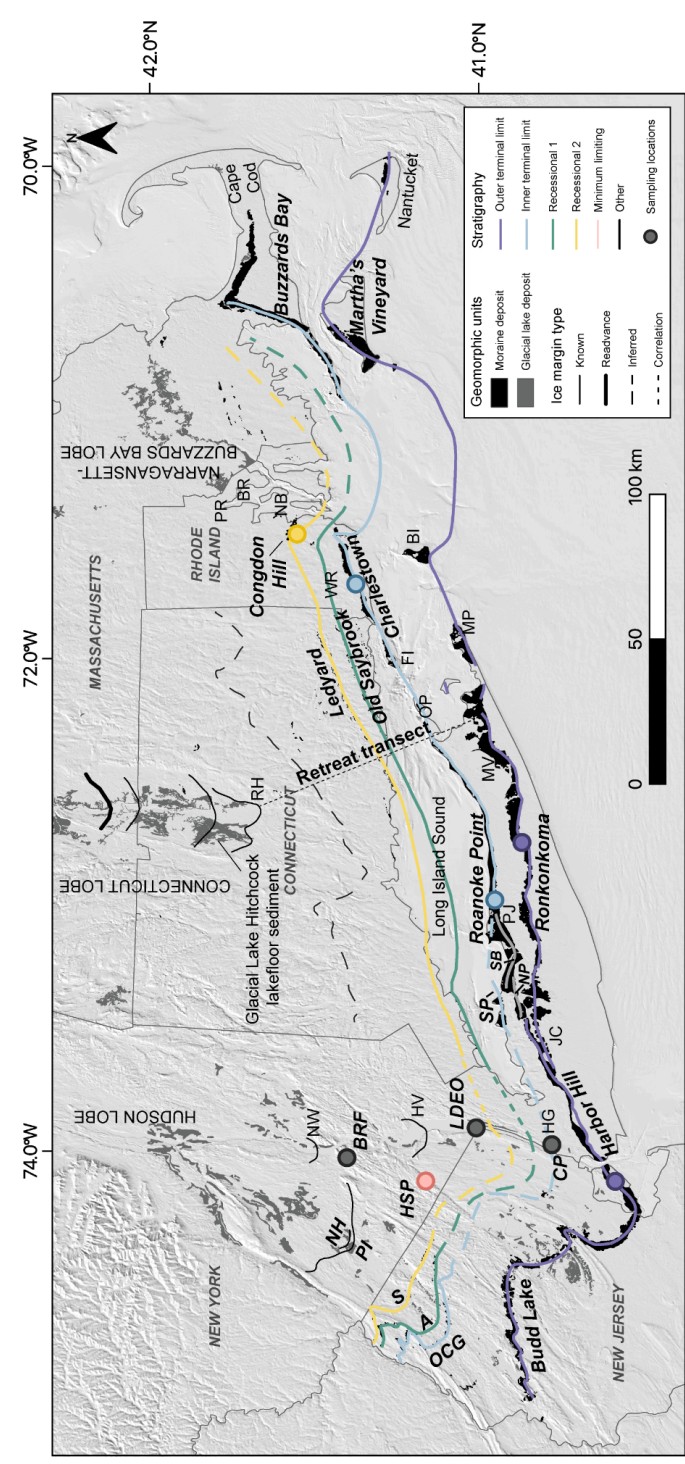






**Figure 1** - Regional map of New England and New York depicting ice marginal positions and glacial geomorphology. Hillshade topography from NASA Shuttle Radar Topography Mission (2013). Bathymetry from NOAA Office of Coast Survey BlueTopo product. Glacial geology is from the surficial geologic maps of Massachusetts (Stone et al., 2018), Rhode Island (Boothroyd et al., 2003), Connecticut (Stone et al., 2005), New York (Cadwell et al., 1989), and New Jersey (Stone et al., 2002). Ice marginal positions and correlations are adapted from Sirkin (1982), Stone and Borns (1986), Boothroyd et al. (1998), Stone et al. (2005), Ridge et al. (2004), Ridge et al. (2012), and Stanford et al. (2021). Retreat rates presented in Section 5.1.3 are calculated using distance along the retreat transect. Moraine segment names discussed in the text are labeled in bold italics and other locations of relevance are labeled in regular text. Sample locations associated with a specific ice-margin position discussed in the text are colored by their stratigraphy as defined in the legend. A = Augusta moraine, BI = Block Island, BR = Barrington, RI, BRF = Black Rock Forest, CP = Central Park, FI = Fishers Island, HG = Hell Gate, HSP = Harriman State Park , HV = Haverstraw, NY, JC = Jericho, NY, LDEO = Lamont-Doherty Earth Observatory, MP = Montauk Point, MV = Manorville, NY, NB = Newburgh, NY, NH = New Hampton moraine, NP = Northport moraine, OCG = Ogdensburg-Culvers Gap moraine, OP = Orient Point, PI = Pellets Island moraine, PJ = Port Jefferson, NY, RH = Rocky Hill, CT, S = Sussex moraine, SB = Stony Brook moraine, SP = Sands Point moraine, WR = Wolf Rocks Moraine.

Ice-margin positions north of the terminal complex are marked by smaller moraines and other ice-marginal
landforms that are different in character from the large coastal moraines. Several discontinuous (individual segments
up to 3 km in length), boulder-rich moraines, interpreted as recording brief readvances or standstills as the Connecticut
and Narragansett-Buzzards Bay Lobes retreated northward from the coastal zone (Stone et al., 2005). These include
the Old Saybrook and Ledyard moraines in Connecticut, which are correlated with the Wolf Rocks and Congdon Hill
moraines in Rhode Island, respectively (Figure 1; Boothroyd et al., 1998; Stone et al., 2005). The boulder-lag nature
of these moraines indicates that they were affected by meltwater near the ice margin (Stone et al., 2005). Based on
their morphostratigraphy, the Connecticut moraines are also tentatively correlated with the Augusta and Sussex
recessional moraines in northern New Jersey (Stone and Borns, 1986; Stone et al., 2005; Figure 1). Ice-marginal
positions without a moraine are marked by the collapsed ice-contact slopes of individual morphosequences deposited
during deglaciation. These features mark the retreat of the ice margin in southern New England and, while they cannot
be correlated across valleys for more regional ice positions, they depict systematic retreat of an active ice margin
(Koteff and Pessl, 1981; Stone et al., 2005).
To summarize, two large end moraine systems comprise a terminal complex that formed during ice-margin
advances as the LIS fluctuated near its maximum extent. The outermost ridges of this complex from northern New
Jersey to Nantucket mark the southernmost extent of the LIS, and the innermost ridges of the terminal complex are
mapped from the north shore of Long Island to Cape Cod and may be correlated with the Ogdensburg-Culvers Gap
moraine in northern New Jersey. Recessional limits in Connecticut and Rhode Island are marked by smaller,
discontinuous moraine segments that are starkly different in nature from the moraines of the terminal complex, and
which may correlate with recessional moraines north of the Ogdensburg-Culvers Gap moraine in New Jersey.





### 1.1.2 Existing chronologic constraints

Numerous studies have used cosmogenic-exposure dating, radiocarbon dating, optically stimulated luminescence (OSL) dating, and glacial lake sediment records to develop deglaciation histories for the LIS in southern New England, New York, and New Jersey (e.g., Dalton et al., 2020; Halsted et al., 2022; Peteet et al., 2012; Ridge, 2004; Ridge et al., 2012; Stone and Borns, 1986). The timing of moraine deposition is constrained by bracketing radiocarbon ($^{14}$C) ages in pre- and post-glacial sediment (e.g., Stanford et al., 2021; Stone and Borns, 1986; Stone et al., 2005), which we have recalibrated to calendar years BP using the INTCal20 database and CALIB 8.2 (Figure 2; Reimer et al., 2020). Moraines and other ice-marginal deposits dammed lakes fed by glacial melt throughout the region, including Lake Albany, which occupied what is now the Hudson River Valley; glacial Lake Hitchcock, in



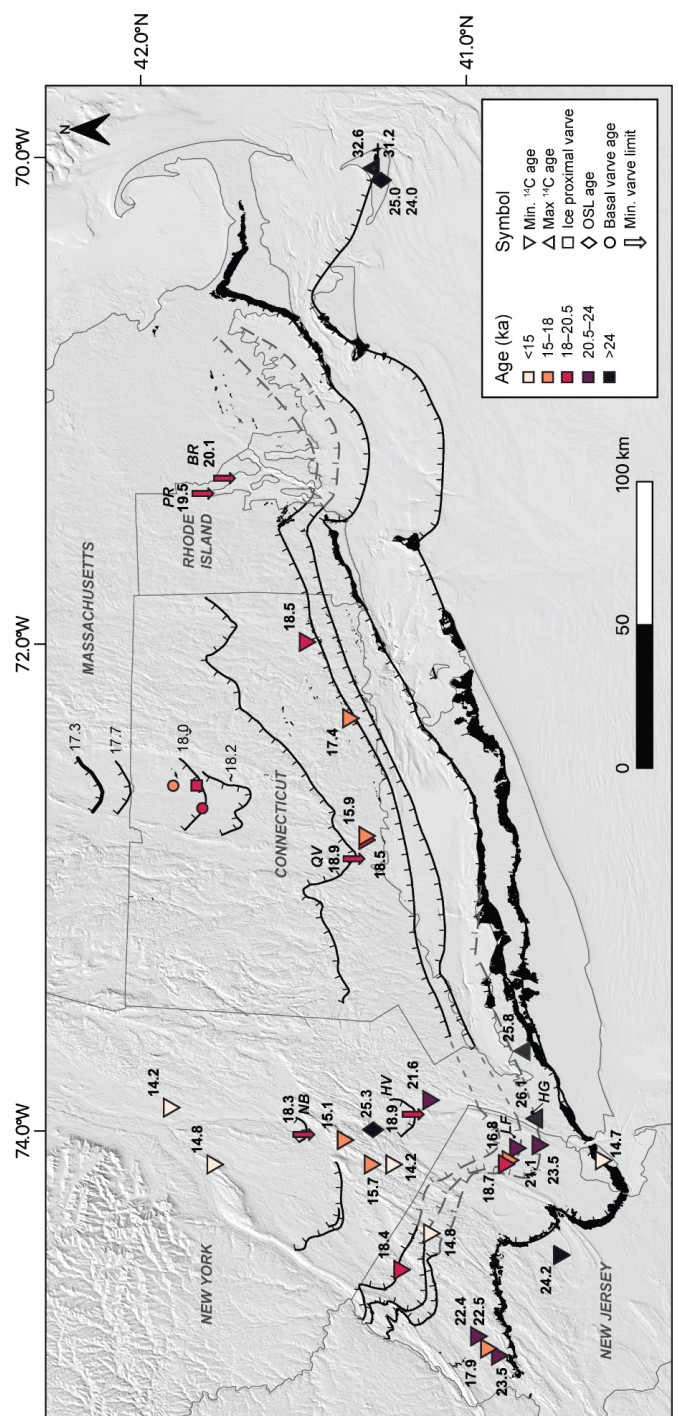

**Figure 2 -** Previously published chronological constraints based on radiocarbon and glacial varves. Background same as Figure 1. Ice margin limit symbols same as Figure 1, but colored in black for simplicity. Ages are discussed and cited in the text. BR = Barrington, RI; HG = Hell Gate; HV = Haverstraw, NY; LF = Little Ferry varve sequence; NB = Newburgh, NY; QV = Quinnipiac Valley, CT; PR = Providence River.



144 what is now the Connecticut River Valley; and lake Narragansett in the Narragansett Bay, Rhode Island (e.g., Antevs,

145 1928, 1922; Oakley and Boothroyd, 2013; Ridge, 2004; Ridge et al., 2012). Annually layered, or varved, sediment

146 throughout the northeast can be aligned across sites to form long varve sequences because the character and thickness

147 of varves deposited in a single year are related to climatic conditions (Antevs, 1928, 1922). These sequences yield a

148 precise chronology of ice margin retreat because (i) the presence of varves indicates ice-free conditions at a given

149 location and (ii) in some cases, a single varve can be traced across sections to its northernmost occurrence, affording

150 a maximum ice margin position for that varve year. The North American Varve Chronology (NAVC) records 5659

151 years of sediment deposition in glacial lakes in New York and New England, including Lakes Hitchcock and Albany,

152 making it the most precise and continuous terrestrial record of LIS retreat through the northeastern United States

153 (Ridge et al., 2012). Varve sequences inboard of LIS moraines also provide minimum limiting ages for those moraines.

154 The NAVC is reported in 'North American Varve Years' numbered 2701-8459, which are calibrated to calendar years

155 by radiocarbon dating of plant macrofossils and other organic material from 54 individual varves throughout the

156 chronology (Ridge et al., 2012). We report NAVC ages in years BP on the Greenland Ice Core timescale (GICC05 yr

157 BP; Andersen et al., 2006) using the offset of 20,925 years (i.e., varve year "0" equals 20,925 years BP) reported in

158 Balco et al. (2021).

159  Absolute ages have been assigned to some of the moraines using cosmogenic exposure dating (Figure 3;

160 Balco et al., 2002; Balco and Schaefer, 2006; Corbett et al., 2017). To integrate the latest developments in cosmogenic-

161 nuclide dating, and to maintain consistency with our new results in this paper, we recalculate published exposure ages

162 using v3 of the online calculator described by Balco et al. (2008), the primary production rate calibration datasets of

163 Borchers et al. (2016) and the scaling method of Lifton et al. (2014; see Methods for further discussion of production

164 rate and scaling method selection). The ages recalculated here therefore differ slightly from the originally reported

165 exposure ages from the same samples given that many of the original publications predate these updated production

166 rate calibration and scaling methods. Finally, we note that while radiocarbon ages and varve years are referenced to

167 1950 CE, the exposure ages are referenced to the time of sample collection (1995–2019 CE). This difference in

168 reference year, however, is negligible for the exposure ages discussed here, which are >18 ka.




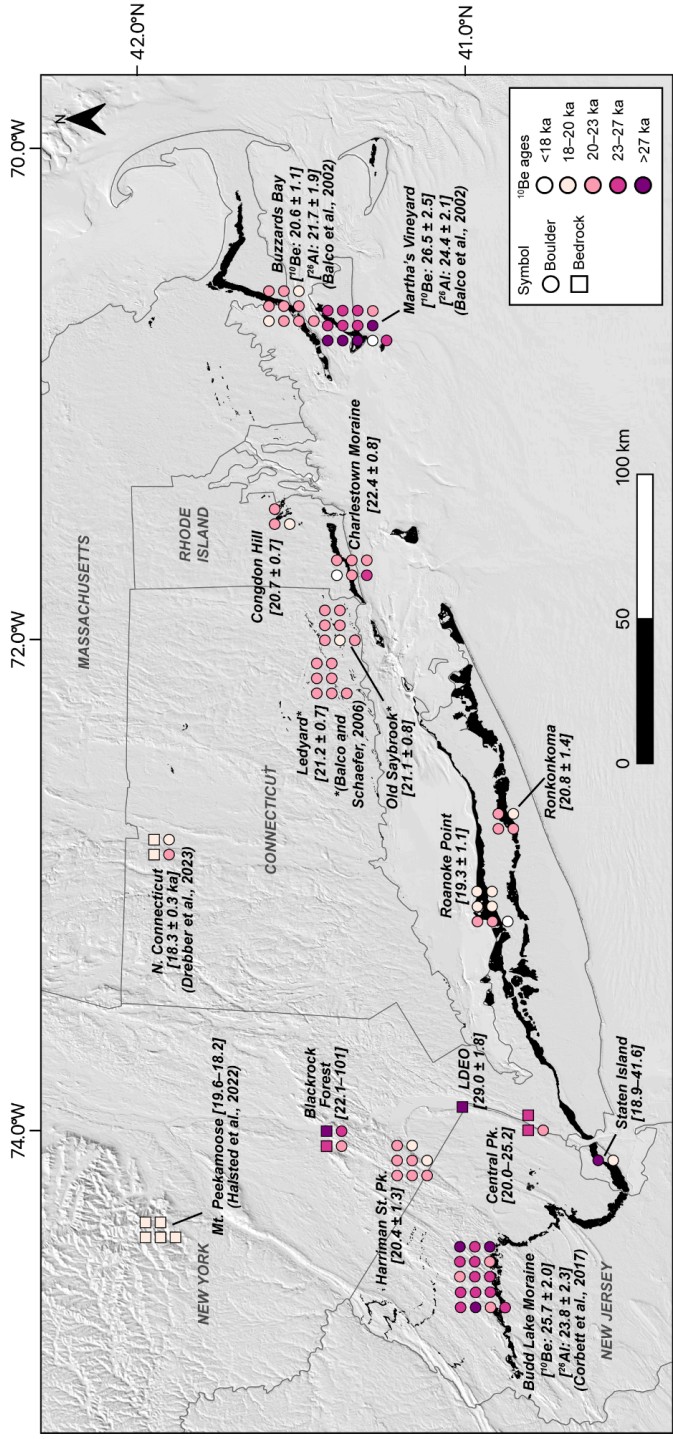

**Figure 3** - New and previously published ¹⁰Be exposure ages from boulder and bedrock surfaces. Background same as in Figure 1. Previously published ages are listed with their reference. All are ¹⁰Be ages, except where ¹⁰Be and ²⁶Al ages are specified. On the Martha's Vineyard and Buzzards Bay moraines, samples with both ¹⁰Be and ²⁶Al measurements are colored according to the average of the ¹⁰Be and ²⁶Al ages (Balco et al., 2002). Although the Budd Lake moraine samples have both ¹⁰Be and ²⁶Al measurements, the symbols are colored only by ¹⁰Be age because Corbett et al. (2017) state that many of the ²⁷Al concentrations may be underestimated and therefore exclude the ²⁶Al ages from discussion. The average of the ²⁶Al ages on the Budd Lake moraine is listed for completeness. Where all samples come from the same deposit, the age is listed as mean ± standard deviation, and where samples at a site are not from the same deposit an age range is listed. A full list of sample ages is in Table 1 and moraine ages in Table 2.

Jo





**Connecticut and Narragansett-Buzzards Bay Lobes:** Existing radiocarbon and exposure ages constrain the
occupation of the outer terminal limit for the Connecticut and Buzzards Bay-Narragansett Lobes to ~27–25 ka.
Maximum limiting radiocarbon ages in preglacial deposits near Boston and on Nantucket suggest that the LIS achieved
its LGM extent in the east by 32–25 ka (29–21 [14]C kyr BP; n = 3; Figure 2; Oldale, 1982; Schafer and Hartshorn,
1965; Tucholke and Hollister, 1973), which agrees with optically stimulated luminescence (OSL) ages from Nantucket
of 24.0 ± 0.9 ka on the oldest moraine and 25.0 ± 0.9 ka on the outboard outwash plain  (Stone and Stone, 2019;
Rittenour, Stone and Mahan, 2012).

[10]Be and [26]Al ages on the Martha's Vineyard moraine range from 17.5 to 63.5 ka (n = 12) and from 17.5 to

60.5 ka (n = 13), respectively (Figure 3; Balco et al., 2002). Some of these exposure ages, especially those older than
the main age population (>30 ka), likely contain [10]Be and [26]Al inherited from a previous exposure period (Balco et al.,
2002). Production of [10]Be and [26]Al  attenuates exponentially with depth in rock (Lal, 1991), meaning that subglacial
erosion of a few meters during glacial periods strips the surface of cosmogenic nuclides that accumulated during prior
exposure (Harbor et al., 2006). Inherited cosmogenic nuclides therefore persist in places where subglacial erosion is
insufficient to remove the signature of prior exposure because of minimally erosive (e.g., cold-based) ice, short ice-
cover durations, or both (e.g., Briner et al., 2006; Stone at al., 2003; Young et al., 2016). The LIS likely remobilized
boulders with significant cosmogenic-nuclide inventories at or near the terminal position as it advanced towards its
LGM extent, so it is not surprising that some of the exposure ages on the terminal moraine are older than its true
emplacement age. Large end moraines with kame and kettle topography, such as the Martha's Vineyard moraine, also
experience permafrost disturbance, which may expose boulders that were previously embedded in the moraine and
shielded from the cosmic-ray flux for some time after deposition (Applegate et al., 2010), or shift or rotate boulders
so original (upon deposition) top surfaces were not sampled. Exposure ages on exhumed or disturbed (e.g., by
agricultural practices and other human activities) boulders are therefore younger than the true emplacement age of the
moraine. Excluding exposure ages likely affected by nuclide inheritance or postdepositional disturbance (n = 4), [10]Be
ages on the Martha's Vineyard moraine average 26.5 ± 2.5 ka (n = 8; mean ± standard deviation) and [26]Al ages average
24.4 ± 2.1 ka (n = 9; Figure 3; Balco et al., 2002; Balco, 2011), and are generally consistent with the maximum limiting
radiocarbon ages in the region and the OSL ages on Nantucket.

[10]Be exposure ages on the Buzzards Bay moraine average 20.6 ± 1.1 ka (n = 10) and [26]Al ages average 21.7

± 1.9 (n = 10; Balco et al., 2002). The Old Saybrook and Ledyard moraines in Connecticut have [10]Be exposure ages
of 21.1 ± 0.8 (n = 7) and 21.2 ± 0.7 ka (n = 7), respectively (Figure 3; Balco and Schaefer, 2006). Thus, although the
moraines represent a recessional sequence and were not deposited simultaneously, their ages are indistinguishable
within 1σ uncertainty. Postglacial sediment containing tundra vegetation at Cedar Swamp, immediately north of the
Ledyard moraine, gives a minimum limiting age for the recessional moraine sequence of 18.5 ± 0.7 ka (mean age ±
2σ uncertainty; 15.2 ± 0.3 [14]C kyr BP; McWeeney, 1995; Stone et al., 2005). A radiocarbon age of 18.5 ± 0.3 ka (15.1
± 0.2 [14]C kyr) at Totocket, near New Haven, Connecticut, also provides a minimum limiting age for the Ledyard
moraine (Figure 2; Davis et al., 1980; Deevey, 1958).

Varve sequences in the region also place minimum age constraints on the recessional moraine sequence. The

NAVC in the Connecticut River Valley begins a few kilometers north of Rocky Hill, the spillway for glacial Lake



Hitchcock (Figures 1 and 2; Antevs, 1928; Ridge et al., 2012). The Rocky Hill sequence overlaps with a varve section
in Newburgh, New York, which together imply that the ice margin had retreated to Newburgh and Rocky Hill by
~18.2 ka (varve year 2701; Figure 2; Antevs, 1928, 1922; Balco et al., 2021; Ridge, 2004; Ridge et al., 2012). Several
varve sections south of Rocky Hill and Newburgh cannot be correlated with the NAVC and are therefore presumed
older, providing minimum estimates for LIS retreat. A ~500 yr varve sequence in the Quinnipiac Valley, near New
Haven, CT, is correlated with a 700-year sequence in Haverstraw, New York, placing a minimum age for ice-free
conditions at Quinnipiac and Haverstraw of >18.9 ka (varve year 2000; Figure 2; Antevs, 1928; Balco and Schaefer,
2006; Ridge et al., 2012). Further east, in the Providence River, Rhode Island, a 600-year varve sequence cannot be
correlated with the NAVC. Summing the Providence River sequence with several varve sequences in Connecticut and
southern Massachusetts between the base of the NAVC and Providence (including the Quinnipiac/Haverstraw varves),
the ice margin must have retreated past Barrington, Rhode Island by ~20.1 ka and north of Providence by ~19.5 ka
(Figures 1 and 2; Oakley and Boothroyd, 2013). Three cosmogenic exposure ages ~30 km north of the Rocky Hill
Dam average 18.3 ± 0.3 ka, corroborating the deglaciation timing in northern Connecticut (Drebber et al., 2023). The
NAVC reveals that the LIS margin was north of New England by 13.6 ka (Ridge et al., 2012), following relatively
minor advances or stillstands at least in the White Mountains and in Maine (e.g., Borns et al., 2004; Bromley et al.,
2015; Davis et al., 2015; Dorion et al., 2001, Hall et al., 2017; Kaplan, 2007; Koester et al., 2017; Thompson et al.,
2017). The position of the retreating ice margin is also marked by annual DeGeer moraines spaced 100 to 300 m apart
in northern New England (Sinclair, 2018; Todd, 2007;  Wrobleski, 2020).

**Hudson Lobe:** Preglacial deposits in Port Washington, New York, and Manhattan, New York, date to 25.8 ± 1.6 ka
(21.8 ± 0.8 [14]C kyr BP) and 26.1 ± 0.3 (21.7 ± 0.1 [14]C kyr BP), respectively, giving maximum limiting ages for the
Hudson Lobe terminal moraine (Figure 2; Schuldenrein and Aiuvalasit, 2011; Sirkin and Stuckenrath, 1980). This is
in agreement with an OSL age of 25.3 ± 7.4 ka on proglacial deposits in Jones Point, New York, associated with the
advance of the Hudson Lobe at Jones Point, New York (Gorokhovich et al., 2018). A radiocarbon age of 24.3 ± 1.1
ka (20.2 ± 0.5 [14]C kyr BP) from an LGM varve sequence at Great Swamp, New Jersey, provides a minimum limiting
age for the Budd Lake moraine in New Jersey (Figure 2; Reimer, 1984; Stanford et al., 2021). Boulders a few km
inboard of the Budd Lake moraine have [10]Be ages of 25.7 ± 2.0 ka (n = 16) and [26]Al ages of 23.8 ± 2.3 ka (n = 16),
although the original publication excludes the [26]Al ages from discussion given evidence that the [27]Al concentrations
were underestimated (Figure 3; Corbett et al., 2017). Together, the existing chronological constraints suggest that the
Hudson Lobe of the LIS reached its southernmost extent by ~25–26 ka and abandoned that limit by ~24 ka (Corbett
et al., 2017; Stanford et al., 2021).

The varves at Haverstraw, New York, place a minimum limiting age of 18.9 ka on the Ogdensburg-Culver

Gap, Augusta and Sussex recessional moraines in northern New Jersey (Ridge et al., 2012). A floating varve sequence
at Little Ferry in Teterboro, NJ, comprises 1100 glacial varves that must be older than the Haverstraw sequence and
1430 postglacial varves that may overlap with the Haverstraw varves (Antevs, 1928, Stanford et al., 2012). The Little
Ferry varves therefore place a minimum limit of ~20 ka on the Augusta position (18.9 + 1.1 kyr; Figure 2). A recent
compilation of chronologic constraints in northern New Jersey, places the base of the Little Ferry varve sequence at



~23.5 ka based on a nearby radiocarbon age (Stanford et al., 2021). Considering these varve sequences alongside
additional radiocarbon ages in northern New Jersey, Stanford et al. (2021) hypothesize that the Hudson Lobe
abandoned the terminal moraine at ~24 ka and retreated to the position of the Sussex moraine, the innermost of the
northern New Jersey recessional moraines, by ~23.5 ka. Based on their revised chronology, Stanford et al. (2021)
suggest that the Connecticut recessional moraines (Ledyard and Old Saybrook) may correlate with the New Hampton
and Pellets Island moraines in New York, rather than the northern New Jersey recessional sequence.

The earliest post-glacial radiocarbon ages on plant macrofossils in lake and bog sediment throughout the
region date to ~14–18.5 ka (Figure 2; Davis et al., 1980; Deevey, 1958; McWeeney, 1995; Stone et al., 2005; Peteet
et al., 2012). These dates provide further minimum limiting ages for the moraine sequences discussed here. The
abundance of macrofossils dating to ~14–16 ka, in addition to the fact that most ages older than 16 ka come from bulk
sediment that are more likely to contain old carbon, has been used to argue that the LIS abandoned its LGM limit ~14–
16 ka (Peteet et al., 2012), and thus ~8–10 ka later than is indicated by the exposure-age and radiocarbon datasets
presented and compiled here.

## 259 2. Geomorphology and study areas

The Connecticut and Narragansett-Buzzards Bay Lobes exhibit exceptionally well preserved moraines that
afford an opportunity to constrain the regional timing of the LGM and its culmination. Below, we describe the
geomorphic settings and sample locations for 40 new exposure ages from the Connecticut and Narragansett-Buzzards
Bay Lobe in Long Island, New York and Rhode Island, as well as from the Hudson Lobe to the west.

### 264 2.1 Connecticut and Narragansett Lobes

### 265 2.1.1 Long Island, New York

Long Island is a large (~200 km long and 35 km wide), densely populated island in the New York
Metropolitan area that extends from Brooklyn, New York City to its eastern extents at Montauk and Orient Point
(Figure 1). The Island comprises tills and outwash plains associated with the southernmost extent of the LIS at the
LGM, and its topography is defined by several prominent moraine ridges (>60 m relief, at points), including the
Ronkonkoma, Harbor Hill and Roanoke Point moraines (Figure 1; Fuller, 1914; Sirkin, 1982; Sirkin and Stuckenrath,
1980). The Ronkonkoma moraine is the stratigraphically oldest (southernmost) associated with the Connecticut Lobe
of the LIS and extends E-W from the hamlet of Jericho in west-central Long Island to Montauk, forming the South
Fork of Long Island. The moraine ridge comprises discontinuous kame deposits and thrust sheets overlain by thin,
sandy till and bisected by outwash-filled valleys (Cadwell, 1989; Sirkin 1982). The easternmost point of the
Ronkonkoma moraine at Montauk Point is correlated with the outer terminal moraine positions on Block Island,
Martha's Vineyard, and Nantucket (Stone and Borns, 1986; Sirkin, 1976). Although boulders ideal for surface
exposure dating were difficult to locate on the Ronkonkoma moraine, we sampled four medium-sized (~1 m height)
granite boulders near Manorville, NY.

The Harbor Hill moraine was originally mapped as extending from New Jersey to Staten Island and across
the north shore of Long Island, crosscutting the Ronkonkoma moraine near Jericho, New York (Fuller, 1914; Figure



1). Yet, updated models of Long Island glaciation demonstrate that the classical Harbor Hill moraine comprises several
segments deposited asynchronously (Sirkin, 1982; Stone and Borns, 1986). Here, the term Harbor Hill moraine refers
to the segment extending from the confluence with the Ronkonkoma moraine through Staten Island, which represents
the terminal limit of the Hudson Lobe in western Long Island (Figure 1), while the Northport and Stony Brook moraine
segments northeast of the confluence with the Ronkonkma moraine are considered younger positions (Sirkin, 1982;
Stone and Borns 1986). A stratigraphic section in Port Washington, New York, reveals that the Harbor Hill moraine
is characterized by ablation till up to 10 m thick overlying thrust sheets of stratified drift containing dislocated
preglacial deposits, suggesting it formed during a readvance (Mills and Wells, 1974; Oldale and O'Hara, 1984).
Several additional moraine segments are mapped north of the Ronkonkoma ice-margin position in eastern Long Island
(Sirkin 1982; Sirkin, 1998), which are not discussed further here.
The Roanoke Point landform is the innermost Connecticut Lobe moraine on Long Island. It appears to
crosscut the Stony Brook moraine segment near Port Jefferson, New York, extending east to Orient Point, forming
the North Fork of Long Island (Figure 1; Cadwell, 1989; Sirkin, 1982). The moraine consists of till over deformed
outwash (Sirkin, 1982). Glaciotectonic structures within the moraine stratigraphy indicate that the moraine was likely
deposited during a readvance of the ice margin, rather than a representing a standstill (Oldale and O'Hara, 1984;
Boothroyd and Sirkin, 2002). The Roanoke Point moraine is correlated with the Sands Point moraine to the west,
deposited by the Hudson Lobe, and tentatively correlated with the Odgensburg-Culvers Gap moraine in northwest
New Jersey (Figure 1; Section 1.1.1; Stanford, 2010, 1993; Stanford et al., 2021; Stanford and Harper, 1991; Stone et
al., 2002, 1995). We sampled seven large (>1 m tall, with some as tall as 4 m) erratic boulders on the Roanoke Point
moraine in the vicinity of Port Jefferson, New York, near Stony Brook University (Figure 4).
Mapping and sampling of the Long Island moraines was undertaken through the Lamont-Doherty Earth
Observatory Undergraduate Student Summer Intern Program between 2002-2006. Original field observations from
2006 note that one sample, LI-9, is located in a topographic depression, and may have been exhumed or toppled after
deposition. Upon further inspection in 2023, many of the samples collected from the Roanoke Point moraine are
located in topographic low points, and only LI-1 and LI-8 were taken from local high points where boulders were less
likely to have been affected by postdepositional processes.



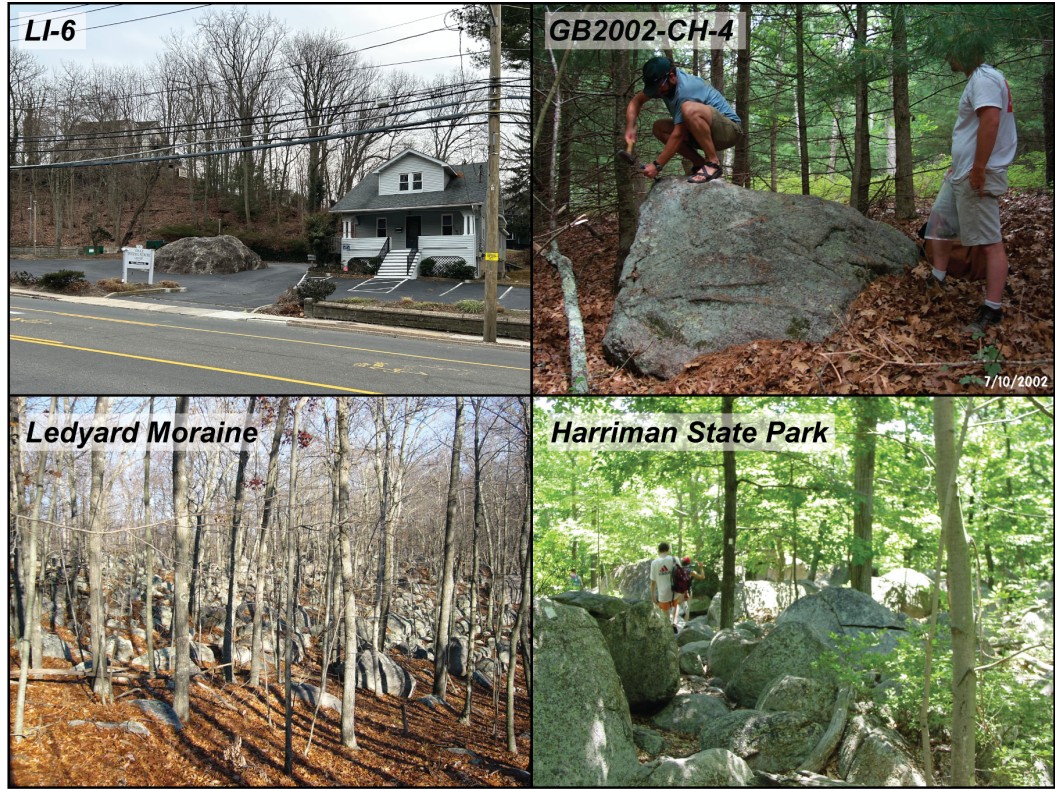

**Figure 4** - Representative sampling locations for surface exposure dating. LI-6: Large boulder sampled in an urban setting of the Roanoke Point moraine on Long Island. The sizable boulder is slightly off the moraine crest (in the background), not located on a local high point and may have experienced postdepositional disturbance. GB2002-CH-4: stable boulder on the Charlestown moraine. Ledyard Moraine: interlocked boulders of the Ledyard moraine in Connecticut. Harriman State Park: Interlocked boulders forming an ice-marginal boulder deposit.


**2.1.2 Rhode Island**
Three ice marginal positions in southern Rhode Island are marked by the Charlestown, Wolf Rocks and
Congdon Hill moraines. The stratigraphically oldest is the Charlestown moraine, which is part of the Roanoke Point
- Fishers Island - Charlestown - Buzzards Bay limit (Figure 1; Kaye, 1960; Upham, 1879). The moraine is ~30 km by
0.5–3 km wide, rising as much as 30 m above the surrounding topography (Kaye, 1960). It is composed of a mixture
of diamict and glaciotectonically displaced stratified deposits (sand and gravel), suggesting it formed during a
readvance, with numerous large boulders at the surface (Boothroyd et al., 1998; Boothroyd et al., 2002; Oldale and
O'Hara, 1984; Schafer, 1965). The Wolf Rocks boulder moraine, which we did not sample, is inboard of the
Charlestown moraine and is correlated with the Old Saybrook recessional moraine in Connecticut (Stone et al., 2005).
The Congdon Hill moraine is the innermost recessional moraine in Rhode Island and is correlated with the Ledyard
recessional moraine in Connecticut to the west (Boothroyd and Sirkin, 2002; Stone et al., 2005). This hummocky



moraine ridge is 6 km long and 3–20 m in height and comprises boulders and sandy till, with numerous large boulders
near the moraine crest (Stone, 2014).

We collected six samples on the Charlestown moraine, and three samples on the Congdon Hill moraine, all

of which were from large (>1 m) boulders (Figure 4). Field observations note that sample GB2002-CH-1 on the
Charlestown moraine was collected from a boulder that had collapsed into a gravel pit. Although it appeared that its
original position could be reconstructed from weathering characteristics and other evidence, this could not be verified.

**2.2 Hudson Lobe**

The Hudson Lobe of the LIS deposited a NE-SW trending moraine on Staten Island that correlates with the

terminal moraines on Long Island to the east (Figure 1; Cadwell, 1989) and in northern New Jersey to the west (Stone,
2002). The hummocky moraine is 0.5–4 km wide by 20 km long, comprising primarily reddish-brown, clayey tills
that are up to ~45 m thick (Soren, 1988). Boulders are rare at the moraine crest (Soren, 1988), but we found two
granite boulders suitable to sample.

We also present new exposure ages from several locations in the Lower Hudson Valley at Central Park in

New York City, Lamont-Doherty Earth Observatory (LDEO), Harriman State Park, and Black Rock Forest. Glacially
molded outcrops of Manhattan Schist in Central Park, New York City, 25 km north of the terminal moraine, are
sparsely overlain by erratic boulders sourced from pegmatitic granites that outcrop ~15 km north of Central Park near
the Bronx Zoo (Brock and Brock, 2001; Jaret et al., 2021; Taterka, 1987). We sampled two quartz veins within
Manhattan Schist, one from Umpire Rock at the southwest corner of Central Park, and one in the northwest corner of
the park, as well as a boulder from the southeast corner of the Sheep Meadow (Collins, 2005). At LDEO, ~50 km
north of the terminal limit, we collected a single sample for surface exposure dating from the Palisades diabase. At
Harriman State Park, ~70 km north of the terminal moraine, we sampled eight large (generally >2 m tall) gneiss or
granitoid boulders from an area with a large concentration of erratics, representing a local ice-marginal deposit, in an
area near two large boulders called the Grandma and Grandpa Rocks (Figure 4). The erratics are perched on bedrock,
or on top of thin till veneer. Finally, at Black Rock Forest, ~90 km north of the outer terminal limit, we collected three
samples of glacially eroded gneissic bedrock and two samples from large (>2 m tall) granite boulders.
**3 Methods**

Samples for surface exposure dating from the upper surfaces of bedrock and erratic boulders were collected

between 2002 and 2006 using the drill-and-blast method of (Kelly, 2003) and/or hammer and chisel. We collected one
replicate sample at Black Rock Forest (BRF-19-01) in 2019. At each site, we measured topographic shielding using a
clinometer and recorded the sample location and elevation using handheld GPS, except for the samples from Rhode
Island for which elevations were measured by barometer traverse from the nearest USGS benchmark. Samples were
processed at the Lamont-Doherty Earth Observatory cosmogenic dating laboratory following established procedures
for isolating quartz and extracting beryllium (e.g., Schaefer et al., 2009). $^{10}$Be/$^9$Be ratios were measured at the Center
for Mass Spectrometry at Lawrence Livermore National Laboratory (LLNL-CAMS) between August 2005 and July





2007, with one additional measurement in July 2019. Prior to 2007, samples were measured relative to the KNSTD standard with a $^{10}$Be/$^9$Be ratio of 3.11 x 10$^{-12}$ (Nishiizumi, 2002). Measurements in 2007 or later were made relative to the 07KNSTD standard with a $^{10}$Be/$^9$Be ratio of 2.85 x 10$^{-12}$ (Nishiizumi et al., 2007), which is taken into account for our $^{10}$Be age calculations (Balco 2008). $^{10}$Be concentrations ranged from 5.61 x 10$^4$ to 6.17 x 10$^5$ with analytical uncertainty of 2–9%. Blank corrections, calculated by subtracting the average number of $^{10}$Be atoms from blanks processed in each sample batch, ranged from <0.5–12%, with the majority of blank corrections being <2% (Table S1). Reported uncertainties in $^{10}$Be concentrations include analytical errors, blank errors, and uncertainties related to the $^9$Be concentration (1.5%) propagated in quadrature.

Apparent $^{10}$Be exposure ages are calculated using Version 3 of the online exposure calculator described by Balco et al. (2008) and subsequently updated, with all information needed to calculate exposure ages available at https://version2.ice-d.org/laurentide/. Here, "apparent" exposure ages refer to the calculated surface age assuming a single period of exposure with no erosion or burial. We note that including the effects of erosion or snow cover would make the ages presented here a few percent older. Since the publication of the first exposure age chronologies in southern New England, efforts have been made to better estimate cosmogenic-nuclide production rates at sites with independent age control (e.g., Balco et al., 2009; Kaplan et al., 2011; Putnam et al., 2019; Young et al., 2013). Of particular relevance here, Balco et al. (2009) established a regional $^{10}$Be production rate calibration dataset for northeastern North America (NENA) using $^{10}$Be measurements at independently dated sites in New England, most of which are part of the NAVC, and on Baffin Island, Canada. In an effort to synthesize several new and existing production rate datasets, Borchers et al. (2016) describe "primary" production rate datasets for $^{10}$Be and $^{26}$Al (among other nuclides), which includes sites that range in latitude and elevation, but does not include calibration sites from NENA. As of this writing, the $^{10}$Be reference production rates calculated using the NENA and Borchers et al. (2016) datasets differ by only ~1.5% (reference production rates calculated in the online production rate calculator described by Balco et al. (2008) and subsequently updated (http://hess.ess.washington.edu/math/v3/v3_cal_in.html; last access January 26, 2023). Given the similarity of these two production-rate datasets, we here employ the $^{10}$Be and $^{26}$Al production rates of Borchers et al. (2016) to avoid circularity when discussing the agreement of the exposure age chronology with the NAVC. In addition, time-dependent production rate scaling frameworks, which account for changes in the geomagnetic field (and therefore cosmic ray flux to the Earth's surface), have been more widely adopted. The LGM moraines discussed here have exposure ages older than those at the production rate calibration sites (Balco et al., 2009; Borchers et al., 2016), so employing time dependent scaling methods may produce more accurate exposure ages. Therefore, we discuss exposure ages calculated using the primary production rate calibration dataset of Borchers et al. (2016) and time-dependent "LSDn" production rate scaling method of Lifton et al. (2014), although also provide ages calculated using the NENA production rate calibration dataset of Balco et al. (2009; NENA) and non-time-dependant "St" scaling (Lal, 1991; Stone, 2000) in Tables S2 and S3. We recognize that the choice of scaling method affects moraine absolute exposure ages by up to ~5% (Table 2), which is within the uncertainty of many moraine ages, but does not affect the calculated rates of net retreat between moraines nor our conclusions.





**4 Results**

Exposure ages from Long Island, New York, and Rhode Island, which are presented in Table 1 and Figure 3,

are relevant to the glacial history of the Connecticut and Narragansett-Buzzards Bay Lobes of the LIS. Ages on the
Ronkonkoma moraine range from 19.1 to 22.4 ka, with an average of 20.8 ± 1.4 ka (average ± SD; n = 4). Boulders
on the Roanoke Point moraine range in age from 18.2 to 20.9 ka, averaging 19.3 ± 1.1 ka (n = 6), with one outlier
that is 14.2 ± 0.6 ka. In Rhode Island, six boulders on the Charlestown moraine have exposure ages that range from
21.8 to 23.7 ka, with one outlier (GB2002-CH-1) excluded because field observations indicated the boulder may not
have been in its original position (Section 2.1.2), as confirmed by an exposure age (17.4 ± 1.6 ka) younger than the
main population of ages on this moraine. The average age of the Charlestown moraine is 22.4 ± 0.8 ka (n = 5).
Boulders on the Congdon Hill moraine range in age from 20.0 to 21.3 ka, and average 20.7 ± 0.7 ka (n = 3).

Additional exposure ages west of Long Island in southern New York, pertain to the Hudson Lobe of the LIS

(Figure 3). On Staten Island, two boulders yield $^{10}$Be ages of 41.6 ± 2.4 and 18.9 ± 2.1 ka. In Central Park, Manhattan,
two $^{10}$Be ages from bedrock samples on Umpire Rock are 25.2 ± 0.8 and 23.2 ± 0.8 and an erratic boulder from Sheep
Meadow yields an age of 20.0 ± 0.7 ka. A single $^{10}$Be age on bedrock at the Lamont-Doherty Earth Observatory is
29.0 ± 1.8 ka. Samples from the ice-marginal deposit in Harriman State Park range in age from 18.7 to 22.8 ka,
averaging 20.4 ± 1.3 ka (n = 8). Finally, three bedrock samples at Black Rock Forest date to 25.0 ± 0.7, 102 ± 3, and
101 ± 3 ka (the latter two bedrock samples are from the same outcrop), and two boulder samples date to 23.7 ± 0.8
and 22.1 ± 0.8 ka.



















**Table 1 -** New [10]Be exposure ages in southern New England and New York. All ages calculated using the primary production rate dataset of Borchers et al. (2016)

| Sample ID | Sample type | [10]Be Age LSDn scaling (yrs) | [10]Be age internal error LSDn Scaling (yrs) | [10]Be Age St scaling (yrs) | [10]Be age internal error St Scaling (yrs) | Included in landform age reported in Table 2? |
|---|---|---|---|---|---|---|
| **Connecticut Lobe** | | | | | | |
| *Ronkonkoma Moraine, Long Island, NY* | | | | | | |
| LI-10 | boulder | 22400 | 800 | 21600 | 700 | yes |
| LI-11 | boulder | 20700 | 700 | 19700 | 600 | yes |
| LI-13 | boulder | 21100 | 700 | 20100 | 700 | yes |
| LI-14 | boulder | 19100 | 800 | 18000 | 700 | yes |
| *Roanoke Point Moraine, Long Island, NY* | | | | | | |
| LI-1 | boulder | 20100 | 1000 | 19100 | 900 | yes |
| LI-3 | boulder | 19800 | 600 | 18800 | 600 | yes |
| LI-4 | boulder | 18300 | 600 | 17300 | 500 | yes |
| LI-6A | boulder | 18900 | 700 | 18000 | 600 | yes |
| LI-6B | boulder | 18300 | 500 | 17300 | 500 | yes |
| LI-7 | boulder | 18200 | 600 | 17200 | 500 | yes |
| LI-8 | boulder | 20900 | 700 | 20000 | 600 | yes |
| LI-9 | boulder | 14200 | 600 | 13300 | 500 | no |





**Table 1 -** Cont'd.

| Sample ID | Sample type | ¹⁰Be Age LSDn scaling (yrs) | ¹⁰Be age internal error LSDn Scaling (yrs) | ¹⁰Be Age St scaling (yrs) | ¹⁰Be age internal error St Scaling (yrs) | Included in landform age reported in Table 2? |
|---|---|---|---|---|---|---|
| *Charlestown Moraine, Rhode Island* | | | | | | |
| GB2002-CH-1 | boulder | 17400 | 1600 | 16500 | 1500 | no |
| GB2002-CH-2 | boulder | 21800 | 800 | 21100 | 700 | yes |
| GB2002-CH-3 | boulder | 22200 | 1100 | 21500 | 1100 | yes |
| GB2002-CH-4 | boulder | 22500 | 800 | 21800 | 700 | yes |
| GB2002-CH-5 | boulder | 23700 | 1000 | 23100 | 1000 | yes |
| GB2002-CH-6 | boulder | 21900 | 1000 | 21100 | 1000 | yes |
| | | | | | | |
| ***Narragansett-Buzzards Bay Lobe*** | | | | | | |
| *Congdon Hill Moraine* | | | | | | |
| GB2002-CO-1 | boulder | 21400 | 700 | 20600 | 700 | yes |
| GB2002-CO-2 | boulder | 20900 | 1000 | 20100 | 1000 | yes |
| GB2002-CO-3 | boulder | 20000 | 1200 | 19100 | 1100 | yes |
| | | | | | | |
| ***Hudson Lobe*** | | | | | | |
| *Harbor Hill Moraine, Staten Island, NY* | | | | | | |
| SI-1 | boulder | 41600 | 2400 | 40700 | 2400 | n/a |
| SI-3 | boulder | 18900 | 2100 | 17800 | 2000 | n/a |



**Table 1 -** Cont'd.

| Sample ID | Sample type | $^{10}$Be Age LSDn scaling (yrs) | $^{10}$Be age internal error LSDn Scaling (yrs) | $^{10}$Be Age St scaling (yrs) | $^{10}$Be age internal error St Scaling (yrs) | Included in landform age reported in Table 2? |
|---|---|---|---|---|---|---|
| *Central Park, Manhattan, NY* | | | | | | |
| UDP-2 | bedrock | 25200 | 800 | 24400 | 700 | n/a |
| UDP-3 | bedrock | 23200 | 800 | 22300 | 800 | n/a |
| UDP-4 | boulder | 20000 | 700 | 19000 | 700 | n/a |
| | | | | | | |
| *Lamont-Doherty Earth Observatory, Palisades, NY* | | | | | | |
| LDEO-1 | bedrock | 29000 | 1800 | 28200 | 1700 | n/a |
| | | | | | | |
| *Harriman State Park, New York* | | | | | | |
| HSP-1 | boulder | 20600 | 700 | 19700 | 700 | yes |
| HSP-2a | boulder | 20300 | 700 | 19400 | 600 | yes |
| HSP-3 | boulder | 21500 | 700 | 20700 | 600 | yes |
| HSP-4 | boulder | 20300 | 800 | 19400 | 700 | yes |
| HSP-06-01 | boulder | 22800 | 800 | 22000 | 800 | yes |
| HSP-06-04 | boulder | 19100 | 700 | 18200 | 700 | yes |
| HSP-06-05 | boulder | 20200 | 600 | 19300 | 600 | yes |
| HSP-06-06 | boulder | 18700 | 700 | 17800 | 700 | yes |



**Table 1 -** Cont'd

| Sample ID | Sample type | ¹⁰Be Age LSDn scaling (yrs) | ¹⁰Be age internal error LSDn Scaling (yrs) | ¹⁰Be Age St scaling (yrs) | ¹⁰Be age internal error St Scaling (yrs) | Included in landform age reported in Table 2? |
|---|---|---|---|---|---|---|
| *Black Rock Forest* | | | | | | |
| BRF-1 | bedrock | 25000 | 700 | 24400 | 600 | n/a |
| BRF-2 | bedrock | 102400 | 2900 | 101400 | 2900 | n/a |
| BRF-3 | boulder | 22100 | 800 | 21500 | 800 | n/a |
| BRF-4 | boulder | 23700 | 800 | 23100 | 800 | n/a |
| BRF-19-01 | bedrock | 101100 | 3000 | 99900 | 3000 | n/a |



## 5 Discussion

The dataset of new and previously reported exposure ages spans the LGM (~26–19 ka), providing insight into the timing of the LIS maximum extent, the LGM duration, and implications for onset of initial retreat in southern New England and New York. We assess the exposure age chronology in more detail to establish an age for each ice limit, present estimates for average retreat rates through the study area and place the moraine chronology in a climatic context.

### 5.1 Moraine ages

### 5.1.1 Connecticut and Narragansett-Buzzards Bay Lobes

The cosmogenic-nuclide chronology for the Connecticut and Narragansett-Buzzards Bay Lobes agrees with limiting age constraints from radiocarbon and glacial lake varves in the region (Figure 2; Figure 5), including for the timing of the LGM and onset of ice recession. The ¹⁰Be (26.5 ± 2.5 ka) and ²⁶Al ages (24.4 ± 2.1 ka) on the Martha's



Vineyard moraine agree within uncertainty with maximum limiting radiocarbon ages in Port Washington, New York
Nantucket, MA, and near Boston, MA, as well as with OSL ages on Nantucket, which together suggest that the
southeastern LIS reached its maximum LGM extent by ~32.4–25.6 ka (Section 1.1.2; Balco et al., 2002; Oldale, 1982;
Rittenour, Stone and Mahan, 2012; Schafer and Hartshorn, 1965; Stone and Stone, 2019; Tucholke and Hollister,
1973). The Ledyard moraine (21.2 ± 0.7 ka; Balco and Schaefer, 2006) and Congdon Hill moraine (20.7 ± 0.7 ka), the
innermost recessional moraines discussed here, are older than minimum limiting ages placed by the varve sequences
in the Quinnipiac Valley (18.9 ka; Ridge et al., 2012) and the Providence River (20.1 ka; Oakley and Boothroyd,
2013).

Average exposure ages for each of the Connecticut and Narragansett-Buzzards Bay moraines are generally
in stratigraphic order, with the terminal limit being ~24.5–26.5 ka, the Roanoke Point-Charlestown-Buzzards Bay
limit being ~19.5–22.5 ka, and the inner limits in Connecticut and Rhode Island being ~20.5–21 ka (Table 2, Figure
6). Upon closer inspection, however, the average exposure ages on the Ronkonkoma moraine (20.8 ± 1.4), Roanoke
Point moraine (19.3 ± 1.1 ka) and Buzzards Bay moraine (21.2 ± 1.6 ka; Balco et al., 2002) are slightly younger than
those of stratigraphically equivalent (Charlestown) and/or inboard (Old Saybrook, Ledyard, and Congdon Hill)
moraines (Figure 6), although the age distributions on equivalent ice-margin limits overlap (Table 3; Figures 7). It is
not required that stratigraphically equivalent moraine segments are exactly the same age, as it is possible that the
timing of moraine emplacement was spatially variable across the region because of long occupation times and/or
asynchronous abandonment of the large moraine belts. Yet, it is expected that outboard moraines are older than those
inboard, so the apparent departure of average moraine age from stratigraphic ordering can be explained if i) the average
ages of the Connecticut and Rhode Island moraines are erroneously old due to nuclide inheritance, and/or ii) the
average ages from the Ronkonkoma, Roanoke Point and Buzzards Bay moraines are spuriously young due to
postdepositional disturbance.

We find it unlikely that the boulders on the Charlestown, Old Saybrook, Ledyard and Congdon Hill moraines
contain significant inherited $^{10}$Be. $^{10}$Be, like most cosmogenic nuclides, is produced by neutron spallation and muon
interactions. Spallation dominates production at the Earth's surface and decreases rapidly with depth at an attenuation
length of ~160 g cm$^{-2}$ at mid-latitudes. Muon interactions account for ~2% of cosmogenic-nuclide production at the
Earth's surface but continue to tens of meters depth in rock, comprising the majority of $^{10}$Be production below ~2 m
depth (Lal, 1991; Balco, 2017). Cosmogenic-nuclide inheritance is most often observed in places where subglacial
erosion is low, such as places with cold-based ice, and is generally more pervasive on bedrock surfaces than boulders
that have been entrained in ice (e.g., Stone et al., 2003; Young et al., 2016). The distribution of boulder exposure ages
on moraines where at least some boulders exhibit inheritance tend to skew old (Applegate et al., 2010), as is the case
on the Martha's Vineyard moraine. The distribution of exposure ages on the Connecticut and Rhode Island moraines,
however, are normal (Table 2; Figure 6), making the presence of inherited spallation-produced $^{10}$Be highly unlikely
in the sampled boulders. Although muon-produced $^{10}$Be accumulates slowly (< 0.1 atom g$^{-1}$ yr$^{-1}$), $^{10}$Be builds to
measurable concentrations even at several meters depth when rock is exposed for the majority of a glacial cycle, as
are landscapes peripheral to the LGM ice sheets. Recent work demonstrates that moraine and erratic boulders near the
LGM limit may therefore contain several-thousand-years' worth of muon-produced $^{10}$Be in excess of the deposition



age even when plucked from rock ~2–6 m below the formerly exposed surface (Briner et al., 2016b; Halsted et al.,
2023). Yet, it is unlikely that all boulders on these moraines, which exhibit an abundance of large boulders (1–2 m;
Figure 4), were sourced from the same depth in

**Figure 5** - Time distance diagram for the Connecticut-Narragansett-Buzzards Bay Lobes of the LIS based on the exposure age, radiocarbon, and varve chronologies. Only $^{10}$Be ages are shown for simplicity. Individual boulder ages are shown as light gray circles and average moraine ages are colored as in Figure 1. Inset shows the slopes associated with various retreat rates.







**Table 2** – Moraine ages and statistics.

| Moraine name | Distance from terminal moraine[1] | Boulder count (samples excluded) | LSDn Exposure age (yrs)[2] | 1σ error in LSDn Exposure age (yr)[2] | St Exposure age (yrs)[2] | 1σ error in St Exposure age (yr)[2] | Coefficient of Variance (%) | Reduced $\chi^2$ | Reference |
|---|---|---|---|---|---|---|---|---|---|
| ***Outer Terminal Limit*** | | | | | | | | | |
| Budd Lake Moraine | 0 | 16 (0) | 25.7 | 2.0 | 24.9 | 2.1 | 8% | 6.14 | Corbett et al., 2017 |
| Ronkonkoma Moraine | 0 | 4 (0) | 20.8 | 1.4 | 19.9 | 1.4 | 7% | 3.36 | This study |
| Martha's Vineyard Moraine | 0 | 8 (4) | 25.4 | 2.5 | 24.9 | 2.6 | 8% | 6.09 | Balco et al., 2002 |
| ***Inner Terminal Limit*** | | | | | | | | | |
| Roanoke Point Moraine | 10 to 25 | 6 (1) | 19.3 | 1.1 | 18.3 | 1.1 | 6% | 3.33 | This study |
| Charlestown Moraine | 28 | 5 (1) | 22.4 | 0.8 | 21.7 | 0.8 | 4% | 0.68 | This study |
| Buzzards Bay Moraine | 8 to 30 | 10 (0) | 21.2 | 1.6 | 20.6 | 1.7 | 8% | 2.00 | Balco et al., 2002 |
| Roanoke Point-Charlestown-Buzzards Bay Combined | 8 to 30 | 12 (11) | 22.2 | 0.8 | 21.6 | 0.8 | 3% | 1.00[4] | Balco et al., 2002 and this study |



**Table 2** – Cont'd.

| Moraine name | Distance from terminal moraine[1] | Boulder count (samples excluded) | LSDn Exposure age (yrs)[2] | 1σ error in LSDn Exposure age (yr)[2] | St Exposure age (yrs)[2] | 1σ error in St Exposure age (yr)[2] | Coefficient of Variance (%) | Reduced χ2 | Reference |
|---|---|---|---|---|---|---|---|---|---|
| *Recessional Limit 1* | | | | | | | | | |
| Old Saybrook Moraine | 35 to 43 | 7 (0) | 21.1 | 0.8 | 20.4 | 0.9 | 4% | 2.10 | Balco and Schaefer, 2006 |
| *Recessional Limit 2* | | | | | | | | | |
| Ledyard Moraine | 44 to 46 | 7 (0) | 21.2 | 0.7 | 20.4 | 0.7 | 3% | 1.21 | Balco and Schaefer, 2006 |
| Congdon Hill Moraine | 50 | 3 (0) | 20.7 | 0.7 | 19.9 | 0.7 | 3% | 0.59 | This study |
| Ledyard-Congdon Hill Combined | 44 to 50 | 10 (0) | 21.0 | 0.7 | 20.2 | 0.7 | 3% | 0.91 | This study |
| *Minimum Limit* | | | | | | | | | |
| Harriman State Park (ice-marginal deposit) | 40 to 50 | 8 (0) | 20.4 | 1.3 | 19.6 | 1.3 | 6% | 2.92 | This study |



[1]Measured parallel to transect in Figure 1.

[2]All ages calculated using the primary production rate dataset of Borchers et al. (2016). Ages calculated using the NENA production rate dataset of Balco et al. (2009) shown in Table S3. All ages are from 10Be, except for on the Martha's Vineyard and Buzzards Bay moraines, for which 26Al and 10Be measurements are reported and discussed in the original publication (Balco et al., 2002). 26Al measurements are also reported for the Budd Lake moraine, but Corbett et al. (2017) do not discuss them because the 27Al concentrations may have been underestimated for at least several samples, so the 26Al exposure ages are not included in the moraine age calculation here.

[3]To calculate this moraine age and statistics, we: include the oldest boulder on the Roanoke Point moraine (LI-8); exclude the youngest four boulders on the Buzzards Bay moraine, as well as sample GB2002-BB2-29-1 because including it raises the reduced $\chi^2$ value to ~40; and exclude the youngest boulder on the Charlestown moraine. Including sample GB2002-BB2-29-1 in the average does not change the rounded exposure age reported here.

[4]Sample GB2002-BB2-29-1 is excluded from the average because including it raises the reduced $\chi^2$ value to ~40. Including this sample does not affect the rounded exposure age reported here.



rock. If some boulders were sourced above this zone, we would expect to see more scatter in these exposure-age
datasets; if at least some boulders were sourced below these depths, inherited muon-produced $^{10}$Be in those samples
would be negligible, and the age distribution would still skew old (Briner et al., 2016b). The morphology of the
moraines along with the uniform age distributions and lack of scatter suggest that the exposure ages on these moraines
represent their true deposition age within uncertainties (Table 2).

.

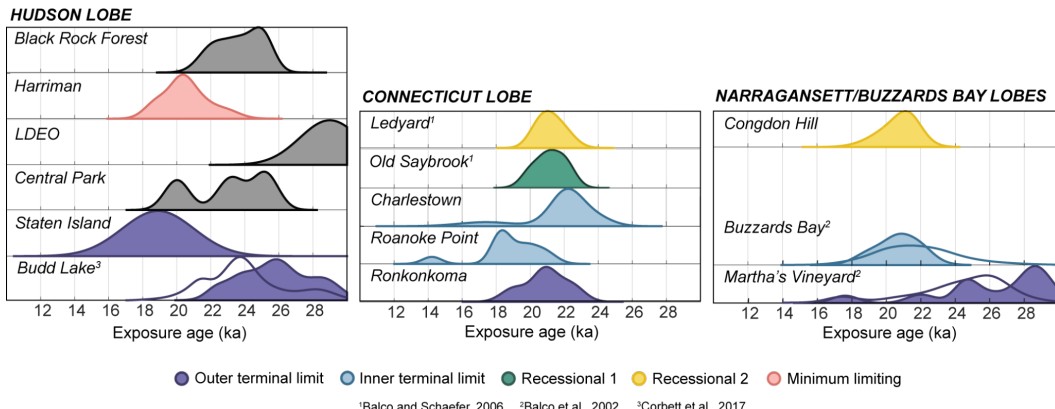

**Figure 6** - Camel plots for moraine exposure ages grouped by LIS lobe. Colors are the same as Figure 1. Filled camel plots show the probability distribution for the $^{10}$Be age of the moraine and open camel plots show the probability distribution for the $^{26}$Al age. Note that normal age distribution of the Ledyard, Old Saybrook, Charlestown and Harriman moraine boulders compared to the age distribution of the Martha's Vineyard moraine, likely reflecting inheritance.


Instead, the preponderance of boulders with ages that may be slightly younger than the true emplacement age
on the Ronkonkoma, Roanoke Point and Buzzards Bay moraines is most likely explained by a small degree of
postdepositional disturbance. These large end moraines have broad, relatively flat crests comprising a complex of
moraine ridges with kettle and kame topography, indicating that the moraines were almost certainly ice cored after
the LIS abandoned these positions and underwent post-emplacement settling. In addition, agricultural disturbances or
other human-induced environmental modification may have contributed to the movement or exhumation of boulders
on these moraines. Balco (2011) recognized that the Buzzards Bay $^{10}$Be and $^{26}$Al measurements, independent
measurements that should be uncorrelated if scatter in the dataset is due to measurement error alone, were in fact
correlated unless the four youngest ages are discarded, indicating the presence of geologic scatter. A moderate
relationship between boulder height and exposure age ($r^2 = 0.36$) suggests that sediment or snow cover is the likely
source of this scatter (Balco, 2011). Discarding the four youngest ages gives an average age of 22.1 ± 0.6 ka for the
Buzzard's Bay moraine. The geomorphic setting of the boulders sampled on the Ronkonkoma and Roanoke Point
moraines indicates a similar role for post depositional disturbance as on the Buzzards Bay moraine. Boulders suitable
for exposure-age dating were difficult to locate on the Ronkonkoma moraine as the moraine comprises mostly sandy
outwash till, which may have been affected by LIS meltwater as it occupied a more northern position (Section 2.1.1).



Samples on the Roanoke Point moraine generally came from large boulders (>1 m) situated in local depressions and/or
inboard of the moraine crest (Section 2.1.1), so may have been

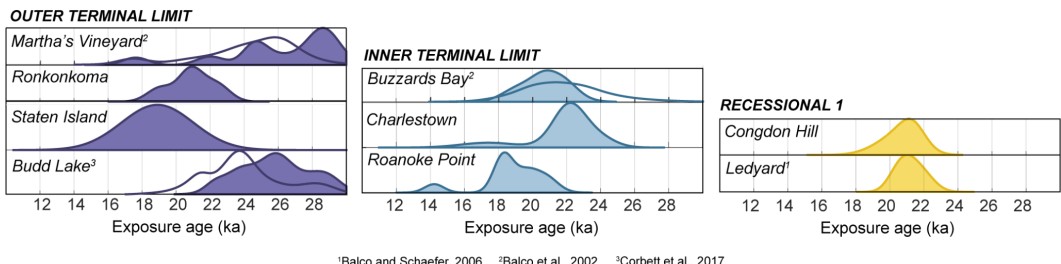

**Figure 7** - Camel plots for moraine exposure ages grouped by ice-margin limit. Colors are the same as Figure 1 and only ice-margin limits with more than one moraine are shown. Filled camel plots show the probability distribution for the [10]Be age of the moraine and open camel plots show the probability distribution for the [26]Al age.

subjected to hillslope processes and/or been encased in stagnant ice even after initial moraine abandonment. It is also
possible these boulders were affected by human modification of the environment. In contrast, the two oldest boulders
on the Roanoke Point moraine (LI-1, $20.1 \pm 1.0$ ka and LI-8, $20.9 \pm 0.7$ ka), while also located slightly inboard of the
moraine crest, rest on local highs where they may have been more stable.
Given the geomorphic context of the Ronkonkoma, Roanoke Point and Buzzards Bay samples, it is possible
that averaging the exposure ages of all boulders from these moraines slightly underestimates the true emplacement
age. On the other hand, the oldest exposure ages from the Ronkonkoma, Roanoke Point and Buzzards Bay moraines
generally overlap with the age distributions of stratigraphically equivalent or inboard moraines (Figures 6 and 7),
suggesting that the oldest ages of the main population are a better estimate of the emplacement age than the average
age. The wide age distribution on the Martha's Vineyard moraine, which includes young ages (~17 ka), is also
consistent with the interpretation that at least parts of the large, hummocky end moraines are affected by
postdepositional disturbance (Figures 6 and 7; Balco et al., 2002). The Martha's Vineyard age distribution also
includes older ages indicative of inheritance, which is to be expected given that the first advance of the LIS to its
terminal position likely remobilized boulders exposed during the preceding interglacial period and prior to expansion
to the southernmost limits.
Guided by these arguments, we present emplacement ages for the moraines deposited by the Connecticut and
Narragansett-Buzzards Bay Lobes of the LIS, recognizing that they are differentially affected by postdepositional
disturbance and nuclide inheritance. For the Martha's Vineyard moraine, we take the average of the [10]Be and [26]Al
ages of the main population, which yields an age of $25.4 \pm 2.5$ ka (Balco et al., 2002). The oldest age on the
Ronkonkoma moraine ($22.4 \pm 0.8$ ka) is probably closer to the true deposition age than the average ($20.8 \pm 1.4$ ka).
For the Roanoke Point-Charlestown-Buzzards Bay limit, we take the average age of the Buzzards Bay boulders,
excluding the four youngest (Balco, 2011); the oldest boulder on Roanoke Point moraine; and the main age population
on Charlestown moraine, which gives an age for this limit of $22.2 \pm 0.8$ ka (Table 2). We take the average age of the
Old Saybrook moraine to represent its true deposition age ($21.1 \pm 0.8$ ka; Balco and Schaefer, 2006). The Ledyard




Moraine (Balco and Schaefer, 2006) and Congdon Hill moraines are stratigraphically correlated, and their exposure
ages agree within measurement uncertainty (reduced $X^2$ of combined population = 1), so we combine their exposure
ages to represent the true age of the limit (21.0 ± 0.8 ka; Table 2).

**5.1.2 Hudson Lobe**

The exposure-age, radiocarbon, and OSL chronologies for LIS retreat in the Hudson River Valley are

generally consistent, although some conflicts remain (Figures 2 and 3). As described in detail in previous studies, the
cosmogenic exposure ages at the Budd Lake moraine (25.7 ± 2.0 ka; Corbett et al., 2017) agree within uncertainty
with the maximum limiting radiocarbon ages in Long Island and in Manhattan (26.1–25.8 ka; Schuldenrein and
Aiuvalasit, 2011; Sirkin and Stuckenrath, 1980), maximum limiting OSL ages at Jones Point, New York (25.3 ± 7.4
ka; Gorokhovich et al., 2018), and a minimum limiting radiocarbon age of 24.2 ± 1.1 ka in a concretion of postglacial
lake sediment just south of the terminal moraine (Stanford et al., 2021). The Budd Lake moraine exposure ages also
overlap with the age distribution on the Martha's Vineyard moraine (Section 5.1.1; Balco et al., 2002; Corbett et al.,
2017). Two boulders on the Harbor Hill moraine on Staten Island, New York, have disparate ages (18.9 ka and 41.6
ka; Table 1), similar to the distribution of ages on Martha's Vineyard, which is affected by inheritance and
postdepositional disturbance. Therefore, we cannot disprove the hypothesis that the moraine on Staten Island was
deposited at the same time as the Budd Lake, Ronkonkoma, and Martha's Vineyard moraines, as the stratigraphic
correlation suggests.

Exposure ages on bedrock surfaces in New York City and the lower Hudson Valley are consistently older

than co-located boulders. Two bedrock ages (25.2 and 23.2 ka) in Central Park, New York are older than a nearby
boulder (20.0 ka); a single bedrock sample at the Lamont-Doherty Earth Observatory dates to 29.0 ka; and at Black
Rock Forest three bedrock ages (one of 25.0 ka and two of ~100 ka) are significantly older than two boulder samples
from the same location (22.1 ka and 23.7 ka; Figure 3). Furthermore, the bedrock ages at LDEO and Black Rock
Forest are older than nearby radiocarbon ages that suggest the ice margin did not retreat north of LDEO until ~22.5
ka and north of Black Rock Forest until ~20–19 ka (Stanford et al., 2021). The fact that bedrock exposure ages
significantly pre-date nearby boulders and radiocarbon ages indicates cosmogenic-nuclide inheritance, implying that
erosion beneath the LIS at these sampling locations was insufficient to remove [10]Be to background levels in bedrock,
perhaps because ice was thin and slow-flowing or because of short ice-cover durations, or both. The three erratic
boulder ages in our Hudson Valley transect do not exhibit a clear trend with distance from the terminal moraine, where
the age in Central Park (20.0 ka) is significantly younger than two ages at Black Rock Forest (22.1 ka and 23.7 ka),
~80 km to the north. Given the presence of inheritance in the bedrock ages and lack of spatial trend in the boulder
ages in the Hudson Valley, we exclude these ages from further discussion here, and identify additional collection of
bedrock and boulder samples in this region as a potential direction for future work.

The average age of the ice-marginal deposit in Harriman State Park (20.4 ± 1.3 ka; Figures 3 and 6) is

consistent with the minimum limiting age of the varve sequence at Haverstraw, New York (18.9 ka; Ridge et al.,
2012), situated a similar distance from the terminal moraine, and is older than the youngest bedrock age on a former
nunatak at Mt. Peekamoose (18.6 ka) ~80 km to the north (Halsted et al., 2022). Finally, the average [10]Be age of the




Harriman State Park boulders of 20.4 ± 1.3 ka is slightly younger than the Ledyard moraine exposure age (21.2 ± 0.7
ka), although the ages overlap within 1σ uncertainty, supporting the correlation of the Augusta and Sussex limits in
northern New Jersey with the Connecticut moraines (Section 1.1.1; Stone et al., 2005). This interpretation, however,
remains in disagreement with recent work that suggests all three moraines in northern New Jersey are ~23.5, and that
the Connecticut moraines may instead correlate with the Pellets Island and New Hampton moraines to the north
(Figure 1; Stanford et al., 2021). Nevertheless, the age of the Harriman State Park ice-marginal deposit agrees with
the majority of bulk radiocarbon ages in northern New Jersey as summarized by Stanford et al. (2021; Figure 2).
**5.1.3 Summary of regional deglaciation chronology**
To summarize the exposure-age chronology, the southeastern LIS occupied the terminal complex from ~26
to 22 ka, with the outermost moraine ridges dating to 25.4 ± 2.5 ka at Martha's Vineyard (Balco et al., 2002) and 25.7
± 2.0 ka at Budd Lake in New Jersey (Corbett et al., 2017). The inner terminal limit at Roanoke Point-Charlestown-
Buzzards Bay, located 10–30 km north of the outer terminal ridge, dates to 22.2 ± 0.8 ka. The fact that the innermost
portion of the terminal complex is nearly 4 kyr younger than the outermost ridges could represent slow, secular retreat
of the ice margin through this period, or the position of the moraines could reflect fluctuations of the ice margin during
the LGM, with the culmination of readvances occurring within the terminal moraine belt. We prefer the latter
interpretation given that the geomorphology of these moraines indicate construction by an advancing LIS and note
that it is unknown how far ice retreated between readvances (Boothroyd et al., 1998; Oldale and O'Hara, 1984).
Irreversible deglaciation began with the abandonment of the inner terminal moraine at ~22 ka, after which
the ice margin did not reoccupy the terminal complex. Ice-margin positions in southern Connecticut and Rhode Island
are marked by smaller, discontinuous, boulder-rich moraines interpreted as recessional limits deposited during brief
re-advances or standstills (Stone et al., 2005). The Old Saybrook moraine, ~40 km inboard of the outer terminal limit,
is 21.1 ± 0.8 ka (Balco and Schaefer, 2006), and the Ledyard-Congdon Hill limit ~45–50 km north of the outer terminal
ridge, is 21.0 ± 0.8 ka. The ice-marginal deposit in Harriman State Park, which is morpho-stratigraphically inboard of
the Ledyard-Congdon Hill limit, is 20.4 ± 1.3 ka. Therefore, the exposure-age chronology presented here spans ~25.5–
20.5 ka. The LIS then retreated to the spillway for glacial Lake Hitchcock in Rocky Hill, CT, ~90–100 km north of
the outer terminal moraine, by ~18.2 ka (Ridge et al., 2012). A lack of extensive end moraine deposits between the
Ledyard-Congdon Hill limit and Rocky Hill, CT signals a shift to more continuous retreat north of our study area.
The positions of the moraines represent net changes in LIS extent from which we estimate average retreat
rates, calculated using the maximum and minimum distance between moraine ridges measured parallel to the transect
in Figure 1, divided by the difference in age established for each limit (Table 2; Figure 8). Although these rates
represent overall northward movement of the ice-margin position (i.e., retreat), they integrate periods of retreat,
advance, and minimal change given that the moraines themselves were formed during readvances or standstills. In
this context, the terminal moraine belt represents several ice-margin fluctuations, with the rate of change in ice-margin
position from the outer terminal to inner terminal limit averaging <5–10 m yr$^{-1}$. Ice then retreated from the inner
terminal position to the Ledyard-Congdon Hill limit at an average rate of ~10–20 m yr$^{-1}$. Further retreat through
southern Connecticut and Rhode Island was interrupted by several standstills or re-advances during which additional



recessional moraines, including the Old Saybrook moraine, were deposited. After abandoning of the Old Saybrook
moraine, the LIS withdrew to Rocky Hill, Connecticut, at an average rate of ~15–25 m yr$^{-1}$ (Ridge et al., 2012). North
of our study area, the NAVC reveals moderate retreat rates of ~30–100 m yr$^{-1}$ during Heinrich Stadial 1 (~18–15 ka),
with an abrupt increase in retreat rate to >300 m yr$^{-1}$ at the onset of the Bølling-Allerød ~15 ka (Figure 8; Ridge et al.,
2012). Similar retreat rates (100-300 m yr$^{-1}$) are implied by DeGeer moraines interpreted to mark the annual retreat of
the ice margin in southern New Hampshire, Maine and Atlantic Canada around 15 ka. (Sinclair et al., 2018; Todd et
al., 2007; Wrobleski, 2020). Cosmogenic-exposure ages from former nunataks that serve as "dipsticks" for LIS
thickness also show moderate thinning through HS1 followed by rapid LIS thinning at the onset of the Bølling (Halsted
et al., 2022).

The regional moraine chronology is remarkably consistent with the varve chronologies, OSL ages, and many

of the radiocarbon ages throughout the region, as discussed above (Figure 5). Yet, the absence of radiocarbon ages on
plant macrofossils between ~26 and 16 ka remains unresolved (Peteet et al., 2012; Figures 2, 3, and 5). This absence
could potentially be explained by i) poor preservation of macrofossils from this time period, ii) landscape instability
and/or sparse vegetation during the LGM and early deglaciation, iii) the delay of widespread organic sediment
deposition until beaver colonies expanded into the region, damming lakes and wetlands (Kaye, 1962), iv) the
predominance of seepage ponds in permeable sand and other ice proximal coarse deposits on end moraines which are
susceptible to periodic drainage, v) difficulty in coring to the till contact in lake sediment affected by postglacial
permafrost and/or vi) persistent lake ice during HS1 (~18–15 ka) summers that precluded organic lake sedimentation.
Further discussion of the ~10 kyr gap between the moraine emplacement age indicated by the exposure-age
chronology and the widespread occurrence of radiocarbon-dated organic material 16 ka is beyond the scope of this
paper.

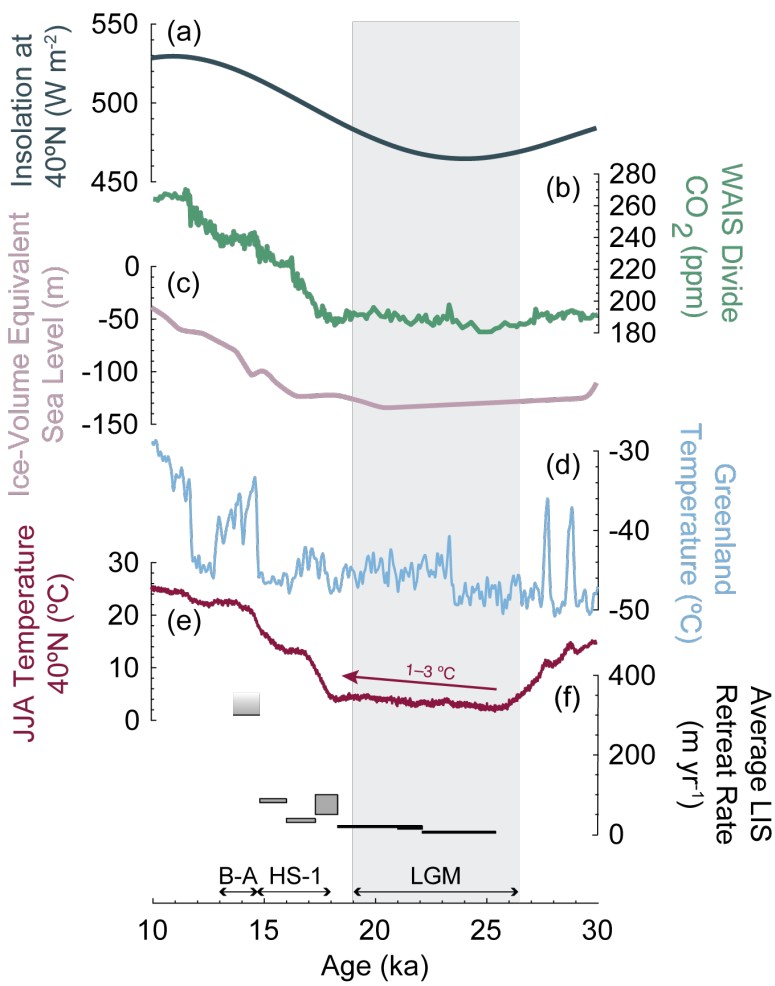

**Figure 8** - LIS ice-margin chronology and average retreat rates compared to other climate parameters and records. a) June 21st insolation at 40ºN (Laskar et al., 2004). b) compilation of atmospheric $CO_2$ measured in Antarctic ice cores (Bereiter et al., 2014; Monnin et al., 2001, 2004; Marcott et al., 2014; Ahn et al., 2014). c) global ice-volume equivalent sea-level (Lambeck et al., 2014). d) Greenland mean-annual temperature reconstruction based on $\partial^{15}N$–$N_2$ in the NGRIP ice core (Kindler et al., 2014). e) time series of summer (June, July and August) surface temperatures modeled using the Had3CMB-M2.1 coupled general circulation model that incorporates Dansgaard-Oeschger and Heinrich events (Armstrong et al., 2019). The time series shown here is for 40ºN, 75.5ºW, ~50 km south of the LGM limit in a part of northern New Jersey that was not covered by ice during the LGM. f) Average LIS retreat rates. Rates shown in black are from this study and those shown in gray are from Ridge et al. (2012), with the faded gray bar indicating a minimum retreat rate of 300 m yr$^{-1}$. The range of average retreat rates are calculated using the maximum and minimum distance between moraine ridges measured parallel to the transect in Figure 1, divided by the difference in age established for each limit (Table 2). Vertical gray bar in the background denotes the LGM timing from 26.5–19 ka (Clark et al., 2009). Heinrich Stadial 1 (HS-1; ~18–15 ka) and Bølling-Allerød (B-A; ~15–13 ka) are periods of abrupt climate change discussed in the text.




### 5.2 Climatic context for initial LIS retreat

The exposure-age-derived moraine chronology suggests that the LIS occupied the terminal moraine complex between ~26 and 22 ka and remained within 50 km of its southernmost position until ~21 ka (Balco et al., 2002; Balco and Schaefer, 2006; Corbett et al., 2017). The moraines discussed here therefore span the canonical LGM and global sea-level lowstand (26.5–19.0 ka; Lambeck et al., 2014; Clarke et al., 2009) and coincide with a local insolation minimum at ~24 ka (Figure 8; Laskar et al., 2004). Furthermore, the timing of terminal moraine occupation from ~26 to 22 ka is similar to that of other LIS sectors to the west, as well as other ice sheets fringing the North Atlantic (Balco et al., 2002; Corbett et al., 2017; Section 5.1.3). For example, exposure and radiocarbon ages indicate the glacial maximum occurred in Wisconsin and Illinois by ~24–23 ka (Ullman et al., 2015; Currey and Petras, 2011) and a minimum limiting radiocarbon age on the terminal moraine of the Miami-Scioto lobe in Indiana and Ohio indicates retreat began sometime before 22.4 ka (Glover et al., 2011). Parts of the British-Irish Ice Sheet began to retreat by ~30–26 ka (Clark et al., 2022) and the Scandinavian ice sheet on Andøya, Norway, fluctuated near its maximum extent from ~26–22 ka (Vorren et al., 2015). Retreat from the terminal moraine complex ~22 ka is consistent with ice-sheet mass balance modeling, which indicates that the moderate increase in local summer insolation beginning ~24 ka may have driven initial LIS margin retreat from its southernmost position (Ullman et al., 2015). We emphasize, however, that the ~50 km of net change in ice-margin position from the outer terminal moraine to the Ledyard-Congdon Hill limit represents <2% of total LIS margin change given that the former LIS is now restricted to the Barnes and Penny Ice Caps on the central Baffin plateau ~3000 km to the north (Dalton et al., 2020; Dyke, 2004; Hooke, 1976; Hooke and Clausen, 1982; Refsnider et al., 2014).

The chronology supports the hypothesis that initial LIS retreat, albeit slow (<5–25 m yr[-1]; Section 5.1.3), began when cold mean-annual temperatures persisted in the Arctic (Kindler et al., 2014) and atmospheric $CO_2$ concentrations remained at glacial values (Figure 8; Denton et al., 2010; Marcott et al., 2014; Raymo, 1997; Ullman et al., 2015; Figure 8). Yet, insight into local summer conditions may provide additional context for the relatively modest LIS retreat during the LGM. Ridge et al. (2012) established a strong relationship, especially after ~15 ka, between LIS retreat rates, the Greenland temperature record, and local summer conditions as recorded by varve thickness, which is largely controlled by LIS meltwater production. In the absence of varve thickness as a proxy for summer climate conditions prior to ~18 ka, we use output from a recent model reconstruction of Northern Hemisphere land surface air temperatures over the last 60 kyr to estimate changes in summer temperature coincident with the moraine chronology discussed here (Figure 8; Armstrong et al., 2019). Modeled terrestrial summer temperature at 40ºN, 75.5ºW, ~50 km south of the LGM limit in northern New Jersey, exhibits a slow but steady increase of ~1–3ºC from 26–19 ka and sharp rise beginning at ~18 ka (Figure 8; Armstrong et al., 2019). The pattern of modeled summer temperature change bears striking resemblance to the slow net LIS retreat (<5–25 m yr[-1]) from ~26–21 ka as indicated by the moraine record, and acceleration of ice-margin retreat after ~18 ka (30–>300 m yr[-1]), as observed in the NAVC (Ridge et al., 2012). We therefore suggest that the relationship between LIS behavior, including relative ice margin positions, and summer conditions observed by Ridge et al. (2012) extends to the LGM. Altogether, the moraine record in southern New England and New York records LIS fluctuations and modest retreat through the LGM, consistent



with a slight increase in modeled summer temperature during that interval, with deglaciation accelerating after 18 ka

alongside the rise in atmospheric $CO_2$ and the onset of Termination 1.

## 6 Conclusions

- The exposure-age chronology in southern New England and New York agrees with established regional stratigraphic relationships and independent age constraints from radiocarbon and glacial lake varves.
- The few inconsistencies in the regional exposure-age dataset can be explained by systematic geologic scatter where i) bedrock samples are affected by nuclide inheritance, ii) the outermost LGM moraine exhibits inheritance on some boulders, and iii) some exposure ages on large unconsolidated landforms that may have experienced extended permafrost conditions are affected by postdepositional disturbance while more stable landforms are not. Also, we cannot rule out that the boulders with the youngest exposure were affected by agricultural practices and other human activities.
- Considering the impact of this geologic scatter, we conclude that the LIS occupied the terminal complex from ~26 ka to ~22 ka (Balco, 2011; Balco et al., 2002). We date several inboard moraines and other recessional deposits to ~21–20.5 ka (Balco and Schaefer, 2006).
- The moraine chronology for the southeastern LIS spans ~26–21 ka, which is consistent with the canonical LGM and sea-level lowstand, full glacial conditions in Greenland, and is broadly coincident with a minimum in local summer insolation.
- Average LIS retreat rates from ~26–18 ka (<5 to 25 m yr$^{-1}$) are consistent with slight warming (1–3ºC) in modeled local summer temperature through the LGM, but were slower than at any point during Termination 1 (>30 to >300 m yr$^{-1}$; Ridge et al., 2012), although this does not account for any distance covered by the readvance or stillstand, if significant. Hence, we conclude that the major pulse of deglaciation and marked recession did not begin until after ~18 ka, when a dramatic rise in atmospheric $CO_2$ signals the onset of Termination 1.

## Data Availability

All analytical information associated with new cosmogenic-nuclide measurements appear in the tables and Supplement. Analytical information, with additional sample documentation and photographs, is also available in the ICED:LAURENTIDE online database (https://version2.ice-d.org/laurentide/, Balco, 2024).

## Competing Interests

The contact author has declared that none of the authors has any competing interests.

## Acknowledgements

We thank the many people who helped support this work in the lab and field, including Sidney Hemming, as well as Mikah McCabe and Rebecca Steinberg who helped collect and process samples as interns in the Lamont-Doherty



Earth Observatory Summer Intern Program. We are immensely grateful to the late Jon Boothroyd of the University
of Rhode Island and the late Gil Hanson of Stony Brook University for sharing their expertise in the regional
stratigraphy and geomorphology. This work was supported in part by the National Science Foundation Graduate
Research Fellowship under grant no. DGE 2036197 to Allie Balter-Kennedy. Joerg Schaefer acknowledges support
by the Vetlesen Foundation and the LDEO Climate Center.

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
