# Peer review of "The Laurentide Ice Sheet in southern New England and New York"

_EGUsphere, 2024_

## Author Response (AR1)

We thank Chris Halsted for his review of the manuscript and insightful comments.

Below, we address referee comments, supplied in bold, with our responses in regular text.

**Line Edits** (indicated by line number)

**Figure 3 – Peekamoose Mt. samples are identified as bedrock, but were actually boulder samples**

Great catch, thank you! Reference to the "Mt. Peekamoose bedrock ages" will be updated to "Peekamoose Mountain boulder ages" in Figure 3 and all instances in the text.

**222 – There is a big jump here between detailed deglacial chronology in southern New England up to ~18 ka and the timing of ice recession from northern New England around 13.5 ka. It might be worth adding a line or two in here to briefly summarize what happened in the intervening 5 kyr. Something as simple as "systematic retreat between 50 – 300 m/yr with relatively minor standstills in northern New Hampshire and Maine around 14 ka" would give the reader just a bit more insight into what happened in the rest of the region.**

We will update these sentences to read:

"The NAVC reveals systematic ice retreat through New England at 50–300 m yr$^{-1}$ (Ridge et al., 2012), with relatively minor advances or stillstands at least in the White Mountains and Maine (e.g., Borns et al., 2004; Bromley et al., 2015; Davis et al., 2015; Dorion et al., 2001, Hall et al., 2017; Kaplan, 2007; Koester et al., 2017; Thompson et al., 2017). The position of the retreating ice margin is also marked by annual DeGeer moraines spaced 100 to 300 m apart in northern New England (Sinclair, 2018; Todd, 2007; Wrobleski, 2020). The LIS margin retreated north of New England by 13.6 ka (Ridge et al., 2012)."

**232 – "Jones Point, New York" is repeated twice in this sentence. Remove the second occurrence.**

We will remove the second occurrence.

**294 – Elaborate what the "Glaciotectonic structures" were that indicate the moraine depositional history**

We will update this sentence to read "Glaciotectonic structures, such as imbricated thrust sheets and dislocated strata, within the moraine stratigraphy indicate that the moraine was likely deposited during a readvance of the ice margin, rather than a representing a standstill (Oldale and O'Hara, 1984; Boothroyd and Sirkin, 2002)."

**Figure 5 – I really like this figure, especially the inset showing retreat rates as slopes, but I wonder if there is a way to identify which specific moraines are represented by the solid symbols. For example, the outer terminal moraine appears to have two solid circles, but in Figure 6 (and in the text), four moraines segments are grouped into the "outer terminal moraine" classification (Martha's Vineyard, Ronkonkoma, Budd Lake, Staten Island). This might not be easily do-able in the figure itself, so maybe providing some more information in the caption would help.**

Figure 5 depicts ages only for the Connecticut-Narragansett-Buzzards Bay Lobes of the LIS, where there are enough dated landforms to estimate retreat rates between limits, while Figure 6 shows the chronology across the entire region separated by ice lobe.

We agree that adding moraine names to Figure 5 will make it easier to follow along with the text. Below each grouping of symbols, we will add shortened versions of the moraine names, in order from oldest to youngest, and update the caption to explain this (see figure in attachment).

**567 – Again, the samples from Peekamoose Mt. were boulders, not bedrock**

Updated!

**Supplementary Tables**

**Table S3:**

- **The header row in this table says "Table S2", check for consistency**
- **Add units to the "Distance from terminal moraine" column**
- **Units for age columns should be kyr, correct?**

Thank you for catching this. We will update Table S3 to include the correct units and table number.

[Figure]

**Figure 5** - Time distance diagram for the Connecticut-Narragansett-Buzzards Bay Lobes of the LIS based on the exposure age, radiocarbon, and varve chronologies. Only [10]Be ages are shown for simplicity. Individual boulder ages are shown as light gray circles and average moraine ages are colored as in Figure 1. Moraine names indicated below each limit in order of oldest to youngest. MV = Martha's Vineyard Moraine, RN = Ronkonkoma Moraine, CH = Charlestown Moraine, BB = Buzzard's Bay Moraine, RP = Roanoke Point Moraine, OS = Old Saybrook Moraine, LD = Ledyard Moraine, CO = Congdon Hill moraine. Inset shows the slopes associated with various retreat rates.

We appreciate Alberto Reyes' review of our manuscript and helpful suggestions.

Below, we address referee comments, supplied in bold, with our responses in regular text. We've also included a description of several unsolicited changes that we plan to make to the manuscript.

**Suggestions and line-corrections follow:**

**line 112: "…are interpreted as…."**

Good catch, we will update this.

**Figs 1,2,3: For those less familiar with the geography in this field area, a blue shading for ocean would be helpful**

We will add blue shading for the ocean in all three figures.

**Fig 2: It is sometimes difficult to quickly link varve/14C locations mentioned explicitly in the main text to this otherwise clear figure. For this map, consider adding more short placename tags to any mapped localities discussed in the main text?**

We agree that several place names mentioned in the main text were not included on this map. We will add the following places to the map and figure caption: Manhattan, New York City (MH), Port Washington, NY (PW), Totoket (TT), Cedar Swamp (CS), Rocky Hill (RH), Jones Point (JO), Great Swamp (GS), and Nantucket. We will also change the abbreviation for Newburgh, NY from NB to NW to avoid confusion with Narraganset Bay, shown as NB in Figure 1.

In addressing this comment, we also noticed a few places missing from Figure 1. We will add Port Washington, NY (PW) and Staten Island (SI). We will also change the abbreviation for Manorville, NY to MN from MV (a common abbreviation for Martha's Vineyard, although not used in the figures), and will update the abbreviation for Newburgh, NY (NW) in the figure caption, which had been misstated as NB (used for Narraganset Bay).

**Fig. 3: I think it actually be easier to assess the data if the coloured ice-margin-positions were retained from Fig 1. I found myself trying to draw them in by hand in colour, to help me compare text to the figure when assessing geomorphic superposition.**

We agree and will add the colored ice margins to Figure 3. We will also retain the color on the ice margins in Figure 2 for consistency. Please see attachment with the new figures and captions.

**lines 228-234: Is it possible to briefly indicate what kind of material is being dated when reporting 14C dates? It's possible this is just opening a can of worms, a detailed treatment of which the authors correctly suggest is beyond the scope of the manuscript.**

We will update the text to specify that the ages discussed in lines 228-234 (as well as the next paragraph) are bulk radiocarbon ages on the described sediment.

**throughout: At several points I found myself wanting to see a more detailed hillshade map of specific sites where morphology of glacial landforms was referred to when**

**justifying rejection of certain dated boulders or when explaining geomorphic/crosscutting relations between landforms (e.g. lines 273-275, 291-294, 312-314, 329-331, 490-493). There's room for this in main text but could also be as supplemental figures. On a related note, I also found myself hoping to see more field photos of boulders/sampling sites, since authors acknowledge that some of these are problematic (more boulder photos can for sure go in supplement).**

We agree that adding more detailed hillshades and photos would aid the discussion of moraine morphology and sample positioning. We will add supplemental figures with hillshade maps in the vicinity of our new sample locations (Long Island, Staten Island, and Southern Rhode Island; see attachment). We are also in the process of also adding additional sample photos to the online database, ICE-D:Laurentide, that hosts our dataset ([https://version2.ice-d.org/laurentide/publication/1187/](https://version2.ice-d.org/laurentide/publication/1187/)). The url is included in the methods section, the Data Availability statement and in the references, and we will update it to point the reader directly to the new samples associated with this manuscript. The photo upload will be complete before publication.

**lines 583-585: Does moraine stratigraphy/sedimentology support this preferred interpretation too?**

Yes, this interpretation is also supported by the sedimentology, as described in Sections 1.1.1 and 2.2.1. We will add this important detail to this sentence to read:

"We prefer the latter interpretation given that the geomorphology and sedimentology of these moraines indicate construction by an advancing LIS and note that it is unknown how far ice retreated between readvances (Boothroyd et al., 1998; Oldale and O'Hara, 1984; Sections 1.1.1 and 2.2.1)."

**lines 644: <2% of margin position change, and probably way less than that when expressed volumetrically…**

Certainly true. We will update the sentence on Lines 642–646 to read:

"We emphasize, however, that the ~50 km of net change in ice-margin position from the outer terminal moraine to the Ledyard-Congdon Hill limit is modest in the context of the entire ice sheet. This distance represents <2% of total LIS margin change considering that the former LIS is now restricted to the Barnes and Penny Ice Caps on the central Baffin plateau ~3000 km to the north (Dalton et al., 2020; Dyke, 2004; Hooke, 1976; Hooke and Clausen, 1982; Refsnider et al., 2014), and likely significantly less when expressed volumetrically since the LIS would have been thinner at its margin than towards the centre of the ice sheet (e.g., Stokes et al., 2012)."

**Unsolicited changes**

1. We will update the label for a boulder photo in Figure 4, which was erroneously labeled LI-6 rather than the correct sample ID, LI-3.
2. We will add a bit of additional information about LIS retreat chronologies in northern New England, in addition to what was requested by Reviewer #1. We will further update the sentences beginning Line 222 to read:

   "The NAVC reveals systematic net ice retreat through New England at 50–300 m yr-1 (Ridge et al., 2012), interrupted by relatively minor advances or stillstands at least in the White Mountains and Maine, documented by comprehensive 14C-based chronologies and 10Be dating (e.g., Borns et al., 2004; Bromley et al., 2015; Davis et al., 2015; Dorion et al., 2001, Hall et al., 2017; Kaplan, 2007; Koester et al., 2017; Thompson et al., 2017). The position of the retreating ice margin is also marked by annual DeGeer moraines spaced 100 to 300 m apart in northern New England (Sinclair, 2018; Todd, 2007; Wrobleski, 2020). The LIS margin had retreated north of New England by 13.6 ka (Ridge et al., 2012) , with slightly later retreat or pockets of smaller residual glaciers perhaps lasting only briefly longer in areas of northern Maine (Borns et al., 2004)."

3. We will correct the age for UDP-2 in Table 1 and throughout the text, which was written as 200 years older than calculated. The age for this sample was already correct in Table S2.
4. An accepted preprint in Climate of the Past suggests that stratigraphic disturbance in lakes affected by postglacial permafrost may be one reason that basal radiocarbon ages older than 16 ka are absent in southern New England. Will include this reference in our brief discussion of the difference between radiocarbon-based and cosmogenic-nuclide-based chronologies in the region near line 671. Point v) will read:

   "difficulty in coring to the till contact and/or stratigraphic disturbance in lake sediment affected by postglacial permafrost (Prince et al., 2024)"

   We currently cite the preprint, but hope the reference can be updated if Prince et al. (2024) is published prior to our manuscript.

5. Finally, we will make small line edits throughout to improve the clarity of the paper.

**Additional References**

[revised manuscript text omitted]

*Correspondence to*: Allie Balter-Kennedy (abalter@ldeo.columbia.edu)

**Contents of this file**
-Figures S1–S3

**Supplemental material uploaded as separate files**
-Tables S1-S3
-Additional sample photos for the new sample locations associated with this publication can be found on ICE-D:Laurentide (https://version2.ice-d.org/laurentide/publication/1187/)

**Figure S1 -** High resolution hillshade of Long Island and representative sample photos from the Roanoke Point and Ronkonkoma moraines. Topography is the New York State 2 m Hillshade from the New York State GIS Program Office (2023). Bathymetry from NOAA Office of Coast Survey BlueTopo product is shown in light grey. Brown moraine outlines are from the surficial geologic map of New York (Cadwell et al., 1989). Photos show representative samples from the Roanoake Point moraine (LI-8, stable position on topographic high; LI-9, boulder located in topographic low, age considered a young outlier) and the Ronkonkoma moraine (LI-13). As described in the text, all samples on the Roanoke point moraine, except LI-1 and LI-8, are in topographic lows, and we therefore hypothesize that they have experienced some degree of postdepositional disturbance, resulting in an average exposure age that is younger than the true moraine deposition age.

[Figure]

**Figure S2 –** Shaded relief map of Rhode Island and representative sample photos from the Charlestown and Congdon Hill moraines. Topography is the Rhode Island Lidar Shaded Relief from Rhode Island Geographic Information System (2022). Bathymetry from NOAA Office of Coast Survey BlueTopo product is shown in light grey. Brown moraine outlines are from the surficial geologic map of Rhode Island (Boothroyd et al., 2003). Photos show representative samples from the Congdon Hill moraine (CO-1, CO-3) and the Charlestown moraine (CH-1, located in gravel pit, age is the one young outlier on this moraine; CH-2).

**Staten Island Sample Locations**

[Figure]

**Figure S3 -** Shaded relief map of Staten Island and sample photos from the Harbor Hill moraine on Staten Island. Topography is the New York State 2 m Hillshade from the New York State GIS Program Office (2023). Bathymetry from NOAA Office of Coast Survey BlueTopo product is shown in light grey. Brown moraine outlines are from the surficial geologic map of New York (Cadwell et al., 1989). Photos show the two samples from the Harbor Hill moraine on Staten Island. SI-1 (41.6 ± 2.4 ka) , is at topographic high, but is considered to have nuclide inheritance, while SI-3 (18.9 ± 2.1 ka) is located in a drainage and is likely affected by postdepositional disturbance.